# Understanding and Improving Ensemble Adversarial Defense

**Yian Deng**
Department of Computer Science
The University of Manchester
Manchester, UK, M13 9PL
yian.deng@manchester.ac.uk

**Tingting Mu**
Department of Computer Science
The University of Manchester
Manchester, UK, M13 9PL
tingting.mu@manchester.ac.uk

## Abstract

The strategy of ensemble has become popular in adversarial defense, which trains multiple base classifiers to defend against adversarial attacks in a cooperative manner. Despite the empirical success, theoretical explanations on why an ensemble of adversarially trained classifiers is more robust than single ones remain unclear. To fill in this gap, we develop a new error theory dedicated to understanding ensemble adversarial defense, demonstrating a provable 0-1 loss reduction on challenging sample sets in adversarial defense scenarios. Guided by this theory, we propose an effective approach to improve ensemble adversarial defense, named interactive global adversarial training (iGAT). The proposal includes (1) a probabilistic distributing rule that selectively allocates to different base classifiers adversarial examples that are globally challenging to the ensemble, and (2) a regularization term to rescue the severest weaknesses of the base classifiers. Being tested over various existing ensemble adversarial defense techniques, iGAT is capable of boosting their performance by up to $17\%$ evaluated using CIFAR10 and CIFAR100 datasets under both white-box and black-box attacks.

## 1   Introduction

Many contemporary machine learning models, particularly the end-to-end ones based on deep neural networks, admit vulnerabilities to small perturbations in the input feature space. For instance, in computer vision applications, a minor change of image pixels computed by an algorithm can manipulate the classification results to produce undesired predictions, but these pixel changes can be imperceptible to human eyes. Such malicious perturbations are referred to as adversarial attacks. These can result in severe incidents, e.g., medical misdiagnose caused by unauthorized perturbations in medical imaging [43], and wrong actions taken by autonomous vehicles caused by crafted traffic images [34].

The capability of a machine learning model to defend adversarial attacks is referred to as adversarial robustness. A formal way to quantify such robustness is through an adversarial risk, which can be intuitively understood as the expectation of the worst-scenario loss computed within a local neighborhood region around a naturally sampled data example [40]. It is formulated as below:

$$R_{adv}(\mathbf{f}) = \mathbb{E}_{(\mathbf{x},y)\sim\mathcal{D}}\left[\max_{\mathbf{z}\in\mathcal{B}(\mathbf{x})} \ell\left(\hat{y}\left(\mathbf{f}(\mathbf{z})\right), y\right)\right], \tag{1}$$

where $\mathcal{B}(\mathbf{x})$ denotes a local neighborhood region around the example $\mathbf{x}$ sampled from a natural data distribution $\mathcal{D}$. The neighbourhood is usually defined based on a selected norm, e.g., a region containing any $\mathbf{z}$ satisfying $\|\mathbf{z} - \mathbf{x}\| \leq \varepsilon$ for a constant $\varepsilon > 0$. In general, $\ell(\hat{y}, y)$ can be any loss function quantifying the difference between the predicted output $\hat{y}$ and the ground-truth output $y$. A

learning process that minimizes the adversarial risk is referred to as adversarially robust learning. In the classification context, a way to empirically approximate the adversarial robustness is through a classification error computed using a set of adversarial examples generated by applying adversarial attacks to the model [4]. Here, an adversarial attack refers to an algorithm that usually builds on an optimization strategy, and it produces examples perturbed by a certain strength so that a machine learning model returns the most erroneous output for these examples [10, 24].

On the defense side, there is a rich amount of techniques proposed to improve the model robustness [10, 1]. However, a robust neural network enhanced by a defense technique would mostly result in a reduced accuracy for classifying the natural examples [52]. A relation between the adversarial robustness and the standard accuracy has been formally proved by Tsipras et al. [38]: *"Any classifier that attains at least $1 - \delta$ standard accuracy on a dataset $\mathcal{D}$ has a robust accuracy at most $\delta p/(1-p)$ against an $L_\infty$-bounded adversary with $\varepsilon \geq 2\eta$."*, where $\varepsilon$ is the $L_\infty$ bound indicating the attack strength, while $\eta$ is sufficiently large and $p \geq 0.5$ and they both are used for data generation. This statement presents a tradeoff between the adversarial robustness and natural accuracy. There has been a consistent effort invested to mitigate such tradeoff by developing defense techniques to improve robust accuracy without sacrificing much the natural accuracy.

Ensemble defense has recently arisen as a new state of the art [4, 25]. The core idea is to train strategically multiple base classifiers to defend the attack, with the underlying motivation of improving the statistical stability and cooperation between base models. Existing effort on ensemble defense has mostly been focused on demonstrating performance success for different algorithmic approaches. It is assumed that *training and combining multiple base models can defend better adversarial attacks as compared to training a single model*. Although being supported by empirical success, there is few research that provides rigorous understanding to why this is the case in general. Existing results on analyzing generalized error for ensemble estimators mostly compare the error of the ensemble and the averaged error of the base models, through, for instance, decomposition strategies that divide the error term into bias, variance, co-variance, noise and/or diversity terms [45, 7, 39]. It is not straightforward to extend such results to compare the error of an ensemble model and the error of a model without an ensemble structure.

To address this gap, we develop a new error theory (Theorem 4.1) dedicated to understanding ensemble adversarial defense. The main challenge in mitigating the tradeoff between the adversarial robustness and natural accuracy comes from the fact that adversarial defence techniques can reduce the classifier's capacity of handling weakly separable example pairs that are close to each other but from different classes. To analyse how ensemble helps address this particular challenge, we derive a provable error reduction by changing from using one neural network to an ensemble of two neural networks through either the average or max combiner of their prediction outputs.

Although ensemble defense can improve the overall adversarial robustness, its base models can still be fooled individually in some input subspaces, due to the nature of the collaborative design. Another contribution we make is the proposal of a simple but effective way to improve each base model by considering the adversarial examples generated by their ensemble accompanied by a regularization term that is designed to recuse the worst base model. We experiment with the proposed enhancement by applying it to improve four state-of-the-art ensemble adversarial defense techniques. Satisfactory performance improvement has been observed when being evaluated using CIFAR-10 and CIFAR-100 datasets [51] under both white-box and black-box attacks.

## 2 Related Work

**Adversarial Attack:** Typical attack techniques include white-box and black-box attacks. The white-box adversary has access to the model, e.g., the model parameters, gradients and formulation, etc, while the black-box adversary has limited knowledge of the model, e.g., knowing only the model output, and therefore is closer to the real-world attack setting [5]. Representative white-box attacks include Deepfool [27] and the projected gradient descent (PGD) [26]. The fast gradient sign method (FGSM) [18] is a simplified one-iteration version of PGD. The momentum iterative method (MIM) [15] improves the FGSM attack by introducing the momentum term. Another two commonly-used and cutting-edge white-box attacks are the Carlini-Wagner (CW) attack [8] that computes the perturbation by optimizing the perturbation scales, and the Jacobian-based saliency map attack (JSMA) [31] that perturbs only one pixel in each iteration. Representative black-box attacks include

the square attack (SA) [3], the SignHunter attack [2] and the simple black-box attack (SimBA) [19], among which SA can generate stronger attacks but is more computationally expensive. A thorough survey on adversarial attacks is provided by Akhtar et al. [1]. AutoAttack [12] encapsulates a selected set of strong white and black-box attacks and is considered as the state-of-the-art attack tool. In general, when evaluating the adversarial robustness of a deep learning model, it should be tested under both the white and black-box attacks.

**Adversarial Defense:** Adversarial training is the most straightforward and commonly used defense technique for improving adversarial robustness. It works by simply expanding the training data with additional adversarial examples. The initial idea was firstly adopted by Christian et al. [11] to train robust neural networks for classification, later on, Madry et al. [26] used adversarial examples generated by the PGD attack for improvement. Overall, this strategy is very adaptive and can be used to defend any attack, but it is on the expense of consuming more training examples. It has been empirically observed that adversarial training can reduce loss curvatures in both the input and model parameter spaces [17, 16], and this reduces the adversarial robustness gap between the training and testing data [52, 47]. These findings motivate the development of a series of regularization techniques for adversarial defense, by explicitly regularizing the curvature or other relevant geometric character-istics of the loss function. Many of these techniques can avoid augmenting the training set and are computationally cheaper. Typical regularization techniques that attempt to flatten the loss surface in the input space include the curvature regularization (CURE) [28], local linearity regularization (LLR) [32], input gradient regularization [29] and Lipschitz regularization [41]. There are also tech-niques to flatten the loss in the model parameter space, such as TRADES [52], misclassification aware adversarial training (MART) [44], robust self-training (RST) [9, 33], adversarial weight perturbation (AWP) [47], and HAT [46], etc. Alternative to adversarial training and regularization, other defense strategies include pruning [35], pre-training [22], feature denoising [48], domain adaptation [36] and ensemble defense [25, 37], etc. Croce et al. [13] has provided a summary of recent advances.

**Ensemble Adversarial Defense:** Recently, ensemble has been actively used in adversarial defense, showing promising results. One core spirit behind ensemble is to encourage diversity between base models in order to achieve an improved ensemble prediction [45, 7, 39]. Therefore, the advances are mostly focused on designing effective ensemble diversity losses in training to improve adversarial robustness. For instance, the adaptive diversity promoting (ADP) method [30] uses Shannon entropy for uncertainty regularization and a geometric diversity for measuring the difference between the predictions made by base classifiers. The transferability reduced smooth (TRS) method [50] for-mulates the diversity term based on cosine similarities between the loss gradients of base models, meanwhile increases the model smoothness via an $l_2$-regularization of these gradients during training. The similar idea of exploiting loss gradients was proposed earlier in the gradient alignment loss (GAL) method [23]. The conditional label dependency learning (CLDL) method [42] is a latest improvement over GAL and TRS, which measures the diversity using both the predictions and loss gradients of the base models. However, the ensemble nature of encouraging diversity can cause vulnerability for some base models over certain subsets of adversarial examples [28]. In practice, this can limit the overall robustness of the ensemble when the other base models are not strong enough to correct the weak ones. To address this, the diversifying vulnerabilities for enhanced robust generation of ensembles (DVERGE) [49] proposes a vulnerability diversity to encourage each base model to be robust particularly to the other base models' weaknesses. The latest development for improving ensemble defense, known as synergy-of-experts (SoE) [14], follows a different research path. For each input, it adaptively selects a base model with the largest confidence to make the final prediction instead of combining all, for which the supporting algorithm and theory have been developed. Some surveys on ensemble adversarial attacks and defense can be found in He et al. [21], Lu et al. [25].

## 3    Notations and Preliminaries

Bold capital and lower-case letters, e.g., $\mathbf{X}$ and $\mathbf{x}$, denote matrices and vectors, respectively, while lower-case letters, e.g., $x$, denote scalars. The $i$-th row and column of a matrix $\mathbf{X}$ are denoted by $\mathbf{x}_i$ and $\mathbf{x}^{(i)}$, respectively, while $x_{i,j}$ and $x_i$ the elements of $\mathbf{X}$ and $\mathbf{x}$. A classification dataset $D = \{(\mathbf{x}_i, y_i)\}_{i=1}^n$ includes $n$ examples, which are referred to as *natural examples*, with $\mathbf{x}_i \in \mathcal{X} \subset \mathbb{R}^d$ (feature vector) and $y_i \in [C] = \{1, 2, \ldots, C\}$ (class label). We sometimes express the label of an example as $y(\mathbf{x})$ or $y_{\mathbf{x}}$. Storing $\mathbf{x}_i$ as a row of $\mathbf{X} \in \mathbb{R}^{n \times d}$ and $y_i$ an element of $\mathbf{y} \in \mathbb{R}^n$, we also denote this dataset by $D = (\mathbf{X}, \mathbf{y})$. The classifier $\mathbf{f} : \mathcal{X} \to [0, 1]^C$ outputs class probabilities

usually computed by a softmax function. Given the computed probability $f_c$ for the $c$-th class, $\hat{y}_{\mathbf{f}}(\mathbf{x}) = \arg\max_{c \in [C]} f_c(\mathbf{x})$ predicts the class. For a neural network, we denote by $\mathbf{W}^{(l)}$ the weight matrix connecting the $l$-th and the $(l-1)$-th layers and by $w_{i,j}^{(l)}$ its $ij$-th element. The $L_2$-norm $\|\cdot\|_2$ is used to compute the vector length, while the $L_\infty$-norm $\|\cdot\|_\infty$ to generate adversarial attacks. Concatenation of two sets is denoted by the symbol $\cup$.

We focus on classification by minimizing a classification loss $\ell(\mathbf{f}(\mathbf{x}), y_{\mathbf{x}})$, and adapt it to $\ell(\mathbf{f}(\mathbf{X}), \mathbf{y})$ for the whole dataset. Also, we use $\ell_{CE}$ to emphasize the cross-entropy loss. The loss gradient is $\nabla\ell(\mathbf{f}(\mathbf{x}), y_{\mathbf{x}}) = \frac{\partial \ell(\mathbf{f}(\mathbf{x}), y_{\mathbf{x}})}{\partial \mathbf{x}}$. A cheap way to estimate the loss curvature is by finite difference approximation [28], e.g., the following curvature measure based on $L_2$-norm:

$$\lambda_{\mathbf{f}}(\mathbf{x}, \boldsymbol{\delta}) = \frac{\|\nabla\ell(\mathbf{f}(\mathbf{x} + \boldsymbol{\delta}), y_{\mathbf{x}}) - \nabla\ell(\mathbf{f}(\mathbf{x}), y_{\mathbf{x}})\|_2}{\|\boldsymbol{\delta}\|_2}, \tag{2}$$

where $\boldsymbol{\delta} \in \mathbb{R}^d$ is a perturbation. It measures how a surface bends at a point by different amounts in different directions. An adversarial example $\tilde{\mathbf{x}} = \phi(\mathbf{f}, \mathbf{x}, A)$ is generated by attacking the classifier $\mathbf{f}$ using an attack algorithm $A$ on a natural example $\mathbf{x}$. It is further adapted to $\tilde{\mathbf{X}} = \phi(\mathbf{f}, \mathbf{X}, A)$ for the set of adversarial examples each generated from a natural example in $\mathbf{X}$. The quantity $\boldsymbol{\delta}(\mathbf{f}, \mathbf{x}, A)$ $= \phi(\mathbf{f}, \mathbf{x}, A) - \mathbf{x}$ is referred to as the *adversarial perturbation* of $\mathbf{x}$, simplified to $\boldsymbol{\delta}_{\mathbf{x}} = \tilde{\mathbf{x}} - \mathbf{x}$. To control the perturbation strength, we restrict $\|\boldsymbol{\delta}_{\mathbf{x}}\|_\infty \leq \varepsilon$ for some $\varepsilon > 0$, which results in the following adversarial example formulation, as

$$\phi_\varepsilon(\mathbf{f}, \mathbf{x}, A) = \min(\max(\phi(\mathbf{f}, \mathbf{x}, A), \mathbf{x} - \varepsilon), \mathbf{x} + \varepsilon), \tag{3}$$

where both $\min(\cdot, \cdot)$ and $\max(\cdot, \cdot)$ are element-wise operators comparing their inputs.

# 4 An Error Theory for Adversarial Ensemble Defense

In adversarial ensemble defense, a widely accepted research hypothesis is that training and combining multiple base classifiers can improve adversarial defense as compared to training a single classifier. However, this hypothesis is mostly supported by empirical successes and there is a lack of formal theoretical justification. In this work, we seek theoretical evidence, proving that, when using multi-layer perceptrons (MLPs) for classification, classification error reduces when applying adversarial defence to the base MLPs of an ensemble as compared to a single MLP, under assumptions feasible in practice. The following theorem formalizes our main result.

**Theorem 4.1.** *Suppose $\mathbf{h}, \mathbf{h}^0, \mathbf{h}^1 \in \mathcal{H} : \mathcal{X} \to [0, 1]^C$ are $C$-class $L$-layer MLPs satisfying Assumption 4.2. Given a dataset $D = \{(\mathbf{x}_i, y_i)\}_{i=1}^n$, construct an ambiguous pair set $A(D)$ by Definition 4.3. Assume $\mathbf{h}, \mathbf{h}^0, \mathbf{h}^1$ are acceptable classifiers for $A(D)$ by Assumption 4.4. Given a classifier $\mathbf{f} \in \mathcal{H} : \mathcal{X} \to \mathbb{R}^C$ and a dataset $D$, assess its classification error by 0-1 loss, as*

$$\hat{\mathcal{R}}_{0/1}(D, \mathbf{f}) = \frac{1}{|D|} \sum_{\mathbf{x} \in D} 1\left[ f_{y_{\mathbf{x}}}(\mathbf{x}) < \max_{c \neq y_{\mathbf{x}}} f_c(\mathbf{x}) \right], \tag{4}$$

*where $1[true] = 1$ while $1[false] = 0$. For an ensemble $\mathbf{h}_e^{(0,1)}$ of two base MLPs $\mathbf{h}^0$ and $\mathbf{h}^1$ through either an average or a max combiner, i.e., $\mathbf{h}_e^{(0,1)} = \frac{1}{2}(\mathbf{h}^0 + \mathbf{h}^1)$ or $\mathbf{h}_e^{(0,1)} = \max(\mathbf{h}^0, \mathbf{h}^1)$, it has a lower empirical 0-1 loss than a single MLP for classifying ambiguous examples, such as*

$$\mathbb{E}_{a \sim A(D)} \mathbb{E}_{\mathbf{h}^0, \mathbf{h}^1 \in \mathcal{H}} \left[ \hat{\mathcal{R}}_{0/1}\left(a, \mathbf{h}_e^{(0,1)}\right) \right] < \mathbb{E}_{a \sim A(D)} \mathbb{E}_{\mathbf{h} \in \mathcal{H}} \left[ \hat{\mathcal{R}}_{0/1}\left(a, \mathbf{h}\right) \right]. \tag{5}$$

We prove the result for MLPs satisfying the following assumption.

**Assumption 4.2** (**MLP Requirement**). Suppose a $C$-class $L$-layer MLP $\mathbf{h} : \mathbb{R}^d \to [0, 1]^C$ expressed iteratively by

$$\mathbf{a}^{(0)}(\mathbf{x}) = \mathbf{x}, \tag{6}$$

$$\mathbf{a}^{(l)}(\mathbf{x}) = \sigma\left(\mathbf{W}^{(l)}\mathbf{a}^{(l-1)}(\mathbf{x})\right), l = 1, 2, ..., L - 1, \tag{7}$$

$$\mathbf{a}^{(L)}(\mathbf{x}) = \mathbf{W}^{(L)}\mathbf{a}^{(L-1)}(\mathbf{x}) = \mathbf{z}(\mathbf{x}), \tag{8}$$

$$\mathbf{h}(\mathbf{x}) = \text{softmax}(\mathbf{z}(\mathbf{x})), \tag{9}$$

where $\sigma(\cdot)$ is the activation function applied element-wise, the representation vector $\mathbf{z}(\mathbf{x}) \in \mathbb{R}^C$ returned by the $L$-th layer is fed into the prediction layer building upon the softmax function. Let $w_{s_{l+1},s_l}^{(l)}$ denote the network weight connecting the $s_l$-th neuron in the $l$-th layer and the $s_{l+1}$-th neuron in the $(l+1)$-th layer for $l \in \{1, 2 \ldots, L\}$. Define a column vector $\mathbf{p}^{(k)}$ with its $i$-th element computed from the neural network weights and activation derivatives, as $p_i^{(k)} = \sum_{s_L} \frac{\partial a_{s_L}^{(L-1)}(\mathbf{x})}{\partial x_k} w_{i,s_L}^{(L)}$ for $k = 1, 2, \ldots d$ and $i = 1, 2, \ldots C$, also a matrix $\mathbf{P_h} = \sum_{k=1}^d \mathbf{p}^{(k)} \mathbf{p}^{(k)T}$ and its factorization $\mathbf{P_h} = \mathbf{M_h} \mathbf{M_h}^T$ with a full-rank factor matrix $\mathbf{M_h}$. For constants $\tilde{\lambda}, B > 0$, suppose the following holds for $\mathbf{h}$:

1. Its cross-entropy loss curvature measured by Eq. (2) satisfies $\lambda_\mathbf{h}(\mathbf{x}, \boldsymbol{\delta}) \leq \tilde{\lambda}$.

2. The factor matrix satisfies $\|\mathbf{M_h}\|_2 \leq B_0$ and $\left\|\mathbf{M_h}^\dagger\right\|_2 \leq B$, where $\|\cdot\|_2$ denotes the vector induced $l_2$-norm for matrix.

We explain the feasibility of the above MLP assumptions in the end of this section.

Although adversarial defense techniques can improve adversarial robustness, new challenges arise in classifying examples that are close to each other but from different classes, due to the flattened loss curvature for reducing the adversarial risk. We refer to a pair of such challenging examples as an *ambiguous pair*. Our strategy of proving improved performance for adversarial defense is to (1) firstly construct a challenging dataset $A(D)$ comprising samples from these pairs, which is referred to as an *ambiguous pair set*, and then (2) prove error reduction over $A(D)$. To start, we provide formal definitions for the ambiguous pair and set.

**Definition 4.3 (Ambiguous Pair).** Given a dataset $D = \{(\mathbf{x}_i, y_i)\}_{i=1}^n$ where $\mathbf{x}_i \in \mathcal{X}$ and $y_i \in [C]$, an *ambiguous pair* contains two examples $a = ((\mathbf{x}_i, y_i), (\mathbf{x}_j, y_j))$ satisfying $y_i \neq y_j$ and

$$\|\mathbf{x}_i - \mathbf{x}_j\|_2 \leq \frac{1}{JB\sqrt{C\left(\tilde{\lambda}^2 - \xi\right)}}, \tag{10}$$

where $J > 2$ is an adjustable control variable, $\tilde{\lambda}$, $B$ and $\xi \leq \tilde{\lambda}^2$ are constants associated with the MLP under Assumption 4.2. The *ambiguous pair set* $A(D)$ contains all the ambiguous pairs existing in $D$, for which $J$ is adjusted such that $A(D) \neq \emptyset$.

In Theorem 4.1, we are only interested in classifiers that do not fail too badly on $A(D)$, e.g., having an accuracy level above 42.5%. Comparing poorly performed classifiers is not very meaningful, also the studied situation is closer to practical setups where the starting classifiers for improvement are somewhat acceptable. Such a preference is formalized by the following assumption:

**Assumption 4.4 (Acceptable Classifier).** Suppose an acceptable classifier $\mathbf{f} : \mathbb{R}^d \to [0,1]^C$ does not perform poorly on the ambiguous pair set $A(D)$ associated with a control variable $J$. This means that, for any pair $a = ((\mathbf{x}_i, y_i), (\mathbf{x}_j, y_j)) \in A(D)$ and for any example $(\mathbf{x}_i, y_i)$ from the pair, the following holds:

1. With a probability $p \geq 42.5\%$, the classifier correctly classifies $(\mathbf{x}_i, y_i)$ by a sufficiently large predicted score, i.e., $f_{y_i}(\mathbf{x}_i) \geq 0.5 + \frac{1}{J}$, while wrongly classifies the other example $\mathbf{x}_j$ to $y_i$ by a less score, i.e., $f_{y_i}(\mathbf{x}_j) \leq 0.5 + \frac{1}{J}$.

2. When the classifier predicts $(\mathbf{x}_i, y_i)$ to class $\hat{y}_i$, the predicted scores for the other classes excluding $y_i$ are sufficiently small, i.e., $f_c(\mathbf{x}_i) \leq \frac{1 - f_{\hat{y}_i}(\mathbf{x}_i)}{C-1}$ for $c \neq y_i, \hat{y}_i$.

Proof for Theorem 4.1 together with a toy illustration example is provided in supplementary material.

**Assumption Discussion.** Assumption 4.2 is feasible in practice. Reduced loss curvature is a natural result from adversarial defense, particularly for adversarial training and regularization based methods [16, 17] as mentioned in Section 2. Regarding its second part determined by neural network weights and activation derivatives, common training practices like weight regularization and normalization help prevent from obtaining overly inflated elements in $\mathbf{P_h}$, and thus bound $\|\mathbf{M_h}\|_2$ and $\left\|\mathbf{M_h}^\dagger\right\|_2$. Following Definition 4.3, the ambiguous pair $a = ((\mathbf{x}_i, y_i), (\mathbf{x}_j, y_j))$ is constructed to

let the classifier struggle with classifying the neighbouring example, e.g., $(\mathbf{x}_j, y_j)$, when it is able to classify successfully, e.g., $(\mathbf{x}_i, y_i)$. Consequently, the success of classifying $(\mathbf{x}_i, y_i)$ is mostly accompanied with a failure of classifying $(\mathbf{x}_j, y_j)$ into $y_i$, and vice versa. In Assumption 4.4, for an acceptable classifier, the first part assumes its failure is fairly mild, while the second part assumes its struggle is between $y_i$ and $y_j$. As shown in our proof of Theorem 4.1, in order for Eq. (5) to hold, a polynomial inequality of the probability $p$ needs to be solved, providing a sufficient condition on achieving a reduced ensemble risk, i.e., $p \geq 42.5\%$. Later, we conduct experiments to examine how well some assumptions are met by adversarially trained classifiers and report the results in supplementary material.

## 5 iGAT: Improving Ensemble Mechanism

Existing ensemble adversarial defense techniques mostly base their design on a framework of combining classification loss and diversity for training. The output of each base classifier contains the probabilities of an example belonging to the $C$ classes. For an input example $\mathbf{x} \in \mathcal{X}$, we denote its output from the $i$-th base classifier by $\mathbf{h}^i(\mathbf{x}) = \left[h_1^i(\mathbf{x}) ..., h_C^i(\mathbf{x})\right]$ for $i \in [N]$, where $N$ denotes the number of used base classifiers. Typical practice for combining base predictions includes the averaging, i.e., $\mathbf{h}(\mathbf{x}) = \frac{1}{N} \sum_{i=1}^{N} \mathbf{h}^i(\mathbf{x})$, or the max operation, i.e., $h_j(\mathbf{x}) = \max_{i \in [N]} \left(h_j^i(\mathbf{x})\right)$. Without loss of generality, we denote the combiner by $\mathbf{h} = \mathbf{c}\left(\mathbf{h}^1, ..., \mathbf{h}^N\right)$. To train the base classifiers, we exemplify an ensemble loss function using one training example $(\mathbf{x}, y_{\mathbf{x}})$, as below

$$L_E(\mathbf{x}, y_{\mathbf{x}}) = \underbrace{\sum_{i=1}^{N} \ell(\mathbf{h}^i(\mathbf{x}), y_{\mathbf{x}})}_{\text{classification loss}} + \omega \text{Reg}\left(\mathbf{h}(\mathbf{x})\right) + \gamma \text{Diversity}(\mathbf{h}^1(\mathbf{x}), \mathbf{h}^2(\mathbf{x}), \ldots, \mathbf{h}^N(\mathbf{x}), y_{\mathbf{x}})), \quad (11)$$

where $\omega, \gamma \geq 0$ are hyperparameters. An example choice for regularization is the Shannon entropy of the ensemble $\mathbf{h}(\mathbf{x})$ [30]. Significant research effort has been invested to diversity design, for which it is optional whether to use the class information in diversity calculation. In the first section of supplementary material, we briefly explain four ensemble adversarial defense techniques highlighting their loss design strategies. These include ADP [30], CLDL [42], DVERGE [49] and SoE [14], and they are used later in Section 6 to test our proposed enhancing approach.

Despite the effort in diversity design that encourages better collaboration between base classifiers, it is unavoidable for some base classifiers to struggle with classifying examples from certain input subspaces. There are intersected subspaces that all the base classifiers are not good at classifying. To address this, we propose an *interactive global adversarial training* (iGAT) approach. It seeks support from adversarial examples globally generated by the ensemble and distributes these examples to base classifiers with a probabilistic strategy empirically proven effective. Additionally, it introduces another regularization term to improve over the severest weakness of the base classifiers. Below we describe our proposal in detail.

### 5.1 Distributing Global Adversarial Examples

We aim at improving adversarial robustness over intersected feature subspaces which are hard for all base classifiers to classify. These regions can be approximated by global adversarial examples generated by applying adversarial attacks to the ensemble, which are

$$(\tilde{\mathbf{X}}, \tilde{\mathbf{y}}) = \left(\phi_\varepsilon(\mathbf{c}\left(\mathbf{h}^1, ..., \mathbf{h}^N\right), \mathbf{X}, A), \mathbf{y}\right), \quad (12)$$

where rows of $\tilde{\mathbf{X}}$ store the feature vectors of the generated adversarial examples. For instance, the FGSM attack can be used as $A$. Instead of feeding the same full set of adversarial examples to train each base classifier, we distribute different examples to different base classifiers, to improve performance and to reduce training time. The generated examples are divided into $N$ groups according to their predicted class probabilities. The $i$-th group $\left(\tilde{\mathbf{X}}^i, \tilde{\mathbf{y}}^i\right)$ is used to train the $i$-th base classifier, contributing to its classification loss.

Our core distributing strategy is to encourage each base classifier to keep improving over regions that they are relatively good at classifying. This design is motivated by our theoretical result. We have proved in Theory 4.1 an error reduction achieved by the ensemble of base MLPs that satisfy

the acceptability Assumption 4.4. This assumption is partially examined by whether the classifier returns a sufficiently high prediction score for the correct class or low scores for most of the incorrect classes for some challenging examples. By keeping assigning each base classifier new challenging examples that they are relatively good at classifying, it encourages Assumption 4.4 to continue to hold. In Section 6.3, we perform ablation studies to compare our proposal with a few other distributing strategies, and the empirical results also verify our design. Driven by this strategy, we propose one hard and one soft distributing rule.

**Hard Distributing Rule:** Given a generated adversarial example $(\tilde{\mathbf{x}}, y) = \left((\tilde{\mathbf{X}})_k, \tilde{y}_k\right)$, the following rule determines which base classifier to assign it:

$$\text{If } h_y^i(\tilde{\mathbf{x}}) > \max_{j \neq i, j \in [N]} h_y^j(\tilde{\mathbf{x}}), \text{assign } (\tilde{\mathbf{x}}, y) \text{ to } \left(\tilde{\mathbf{X}}^i, \tilde{\mathbf{y}}^i\right). \tag{13}$$

We refer to it as a hard distributing rule as it simply assigns examples in a deterministic way. The example is assigned to the base classifier that returns the highest predicted probability on its ground truth class.

**Soft Distributing Rule:** A hard assignment like the above can be sensitive to errors. Alternatively, we propose a soft distributing rule that utilizes the ranking of the base classifiers based on their prediction performance meanwhile introduces uncertainty. It builds upon roulette wheel selection [6], which is a commonly used genetic operator in genetic algorithms for selecting promising candidate solutions. Firstly, we rank in descending order the predicted probabilities $\{h_y^i(\mathbf{x})\}_{i=1}^N$ by all the base classifiers for the ground truth class, and let $r_{\mathbf{x}}(\mathbf{h}^i) \in [N]$ denote the obtained ranking for the $i$-th base classifier. Then, we formulate a ranking-based score for each base classifier as

$$p_i = \frac{2^{N - r_{\mathbf{x}}(\mathbf{h}^i)}}{\sum_{i \in [N]} 2^{i-1}}, \tag{14}$$

and it satisfies $\sum_{i \in [N]} p_i = 1$. A more top ranked base classifier has higher score. Next, according to $\{p_i\}_{i=1}^N$, we apply roulette wheel selection and distribute the example to the selected base classifier. Specifically, the selection algorithm constructs $N$ intervals $\{[a_i, b_i]\}_{i=1}^N$ where $a_1 = 0$, $b_1 = p_1$, also $a_i = b_{i-1}$ and $b_i = a_i + p_i$ for $i = 2, 3, \ldots, N$. After sampling a number $q \in (0, 1]$ following a uniform distribution $q \sim U(0, 1)$, check which interval $q$ belongs to. If $a_i < q \leq b_i$, then the example is used to train the $i$-th base classifier. This enables to assign examples based on ranking but in a probabilistic manner in order to be more robust to errors.

## 5.2 Regularization Against Misclassification

We introduce another regularization term to address the severest weakness, by minimizing the probability score of the most incorrectly predicted class by the most erroneous base classifier. Given an input example $(\mathbf{x}, y_{\mathbf{x}})$, the proposed term is formulated as

$$L_R(\mathbf{x}, y_{\mathbf{x}}) = -\delta_{0/1}\left(\mathbf{c}\left(\mathbf{h}^1(\mathbf{x}), ..., \mathbf{h}^N(\mathbf{x})\right), y_{\mathbf{x}}\right) \log\left(1 - \max_{i=1}^C \max_{j=1}^N h_i^j(\mathbf{x})\right). \tag{15}$$

Here, $\delta_{0/1}(\mathbf{f}, y) \in \{0, 1\}$ is an error function, where if the input classifier $\mathbf{f}$ can predict the correct label $y$, it returns 0, otherwise 1. This design is also motivated by Assumption 4.4, to encourage a weak base classifier to perform less poorly on challenging examples so that its chance of satisfying the acceptability assumption can be increased.

## 5.3 Enhanced Training and Implementation

The proposed enhancement approach iGAT, supported by (1) the global adversarial examples generated and distributed following Section 5.1 and (2) the regularization term proposed in Section 5.2, can be applied to any given ensemble adversarial defense method. We use $L_E$ to denote the original

ensemble loss as in Eq. (11), the enhanced loss for training the base classifiers become

$$\min_{\{\mathbf{h}^i\}_{i=1}^N} \underbrace{\mathbb{E}_{(\mathbf{x},y_\mathbf{x})\sim(\mathbf{X},\mathbf{y})}\left[L_E(\mathbf{x},y_\mathbf{x})\right]}_{\text{original ensemble loss}} + \alpha \underbrace{\sum_{i=1}^N \mathbb{E}_{(\mathbf{x},y_\mathbf{x})\sim(\tilde{\mathbf{X}}^i,\tilde{\mathbf{y}}^i)}\left[\ell_{CE}(\mathbf{h}^i(\mathbf{x}),y_\mathbf{x})\right]}_{\text{added global adversarial loss}} \quad (16)$$

$$+ \beta \underbrace{\mathbb{E}_{(\mathbf{x},y_\mathbf{x})\sim(\mathbf{X},\mathbf{y})\cup(\tilde{\mathbf{X}},\tilde{\mathbf{y}})}\left[L_R(\mathbf{x},y_\mathbf{x})\right]}_{\text{added misclassification regularization}},$$

where $\alpha, \beta \geq 0$ are hyper-parameters. In practice, the base classifiers are firstly trained using an existing ensemble adversarial defense technique of interest, i.e., setting $\alpha = \beta = 0$. If some pre-trained base classifiers are available, they can be directly used instead, and fine-tuned with the complete loss. In our implementation, we employ the PGD attack to generate adversarial training examples, as it is the most commonly used in existing literature and in practice.

## 6    Experiments and Results of iGAT

In the experiments, we compare with six state-of-the-art ensemble adversarial defense techniques including ADP [30], CLDL [42], DVERGE [49], SoE [14], GAL [30] and TRS [50]. The CIFAR-10 and CIFAR-100 datasets are used for evaluation, both containing 50,000 training and 10,000 test images [51]. Overall, ADP, CLDL, DVERGE and SoE appear to be the top performing methods, and we apply iGAT[1] to enhance them. The enhanced, referred to as iGAT$_{\text{ADP}}$, iGAT$_{\text{CLDL}}$, iGAT$_{\text{DVERGE}}$ and iGAT$_{\text{SoE}}$, are compared with their original versions, and additionally GAL [30] and TRS [50].

### 6.1    Experiment Setting

We test against white-box attacks including PGD with 20 inner optimization iterations and CW with $L_\infty$ loss implemented by Wu et al. [47], and the black-box SignHunter (SH) attack [2] with 500 maximum loss queries. In accordance with Carmon et al. [9], the CW attack is applied on 1,000 equidistantly sampled testing examples. We also test against the strongest AutoAttack (AA) [12], which encapsulates variants of the PGD attack and the black-box square attack [3]. All attack methods use the perturbation strength $\varepsilon = 8/255$.

For all the compared methods, an ensemble of $N = 8$ base classifiers with ResNet-20 [20] backbone is experimented, for which results of both the average and max output combiners are reported. To implement the iGAT enhancement, the soft distributing rule from Eq. (14) is used. The two hyper-parameters are set as $\alpha = 0.25$ and $\beta = 0.5$ for SoE, while $\alpha = 5$ and $\beta = 10$ for ADP, CLDL and DVERGE, found by grid search. Here SOE uses a different parameter setting because its loss construction differs from the others, thus it requires a different scale of the parameter range for tuning $\alpha$ and $\beta$. In practice, minor adjustments to hyper-parameters have little impact on the results. The iGAT training uses a batch size of 512, and multi-step leaning rates of $\{0.01, 0.002\}$ for CIFAR10 and $\{0.1, 0.02, 0.004\}$ for CIFAR100. Implementation of existing methods uses either their pre-trained models or their source code for training that are publicly available. Each experimental run used one NVIDIA V100 GPU plus 8 CPU cores.

### 6.2    Result Comparison and Analysis

We compare different defense approaches by reporting their classification accuracies computed using natural images and adversarial examples generated by different attack algorithms, and report the results in Table 1. The proposed enhancement has lifted the performance of ADP and DVERGE to a state-of-the-art level for CIFAR-10 under most of the examined attacks, including both the white-box and black-box ones. The enhanced DVERGE by iGAT has outperformed all the compared methods in most cases for CIFAR-100. In addition, we report in Table 2 the accuracy improvement obtained by iGAT for the studied ensemble defense algorithms, computed as their accuracy difference normalised by the accuracy of the original algorithm. It can be seen that iGAT has positively improved the baseline methods in almost all cases. In many cases, it has achieved an accuracy boost over $10\%$.

---

[1]The source codes and pre-trained models can be found at https://github.com/xqsi/iGAT.

Table 1: Comparison of classification accuracies in percentage reported on natural images and adversarial examples generated by different attack algorithms under $L_\infty$-norm perturbation strength $\varepsilon = 8/255$. The results are averaged over five independent runs. The best performance is highlighted in bold, the 2nd best underlined.

| | | Average Combiner (%) | | | | | Max Combiner (%) | | | | |
|---|---|---|---|---|---|---|---|---|---|---|---|
| | | Natural | PGD | CW | SH | AA | Natural | PGD | CW | SH | AA |
| CIFAR10 | TRS | 83.15 | 12.32 | 10.32 | 39.21 | 9.10 | 82.67 | 11.89 | 10.78 | 37.12 | 7.66 |
| | GAL | 80.85 | 41.72 | 41.20 | 54.94 | 36.76 | 80.65 | 31.95 | 27.80 | 50.68 | 9.26 |
| | SoE | 82.19 | 38.54 | 37.59 | **59.69** | 32.68 | 82.36 | 32.51 | 23.88 | 41.04 | 18.37 |
| | iGAT$_{SoE}$ | 81.05 | 40.58 | 39.65 | 57.91 | 34.50 | 81.19 | 31.98 | 24.01 | 40.67 | 19.65 |
| | CLDL | 84.15 | 45.32 | 41.81 | 55.90 | 37.04 | 83.69 | 39.34 | 32.80 | 51.63 | 15.30 |
| | iGAT$_{CLDL}$ | 85.05 | 45.45 | 42.00 | 58.22 | 37.14 | 83.73 | 40.84 | 34.55 | 51.70 | 17.03 |
| | DVERGE | 85.12 | 41.39 | 43.40 | 57.33 | 39.20 | 84.89 | 41.13 | 39.70 | **54.90** | 35.15 |
| | iGAT$_{DVERGE}$ | **85.48** | 42.53 | 44.50 | 57.77 | 39.48 | **85.27** | **42.04** | **40.70** | 54.79 | **35.71** |
| | ADP | 82.14 | 39.63 | 38.90 | 52.93 | 35.53 | 80.08 | 36.62 | 34.60 | 47.69 | 27.72 |
| | iGAT$_{ADP}$ | 84.96 | **46.27** | **44.90** | 58.90 | **40.36** | 80.72 | 39.37 | 35.00 | 48.36 | 29.83 |
| CIFAR100 | TRS | 58.18 | 10.32 | 10.12 | 15.78 | 6.32 | 57.21 | 9.98 | 9.23 | 14.21 | 4.34 |
| | GAL | 61.72 | 22.04 | 21.60 | 31.97 | 18.01 | 59.39 | 19.30 | 13.60 | 24.73 | 10.36 |
| | CLDL | 58.09 | 18.47 | 18.01 | 29.33 | 15.52 | 55.51 | 18.89 | 13.07 | 22.14 | 4.51 |
| | iGAT$_{CLDL}$ | 59.63 | 18.78 | 18.20 | 29.49 | 14.36 | 56.91 | 20.76 | 14.09 | 20.43 | 5.20 |
| | SoE | 62.60 | 20.54 | 19.60 | **36.35** | 15.90 | 62.62 | 16.00 | 11.40 | 24.25 | 8.62 |
| | iGAT$_{SoE}$ | **63.19** | 21.89 | 19.70 | 35.60 | 16.16 | **63.02** | 16.02 | 11.45 | 23.77 | 8.95 |
| | ADP | 60.46 | 20.97 | 20.55 | 30.26 | 17.37 | 56.20 | 17.86 | 13.70 | 21.40 | 10.03 |
| | iGAT$_{ADP}$ | 60.17 | 22.23 | 20.75 | 30.46 | 17.88 | 56.29 | 17.89 | 14.10 | 21.47 | 10.09 |
| | DVERGE | 63.09 | 20.04 | 20.01 | 32.74 | 17.27 | 61.20 | 20.08 | 15.30 | 27.18 | 12.09 |
| | iGAT$_{DVERGE}$ | 63.14 | **23.20** | **22.50** | 33.56 | **18.59** | 61.54 | **20.38** | **17.80** | **27.88** | **13.89** |

Table 2: Accuracy improvement in percentage by iGAT, i.e. $\frac{\text{iGAT- original}}{\text{original}} \times 100\%$, reported on natural images and adversarial examples generated by different attack algorithms under $L_\infty$-norm perturbation strength $\varepsilon = 8/255$.

| | | Average Combiner | | | | | Max Combiner | | | | |
|---|---|---|---|---|---|---|---|---|---|---|---|
| | | Natural | PGD | CW | SH | AA | Natural | PGD | CW | SH | AA |
| CIFAR10 | ADP | +3.43 | +16.75 | +15.42 | +11.28 | +13.59 | +0.80 | +7.51 | +1.16 | +1.40 | +7.61 |
| | DVERGE | +0.42 | +2.75 | +2.53 | +0.77 | +0.71 | +0.45 | +2.21 | +2.52 | −0.20 | +1.59 |
| | SoE | −1.39 | +5.29 | +5.48 | −2.98 | +5.57 | −1.42 | −1.63 | +0.54 | −0.90 | +6.97 |
| | CLDL | +1.07 | +0.29 | +0.45 | +4.15 | +0.27 | +0.05 | +3.81 | +5.34 | +0.14 | +11.31 |
| CIFAR100 | ADP | −0.48 | +6.01 | +0.97 | +0.66 | +2.94 | +0.16 | +0.17 | +2.92 | +0.33 | +0.60 |
| | DVERGE | +0.08 | +15.77 | +12.44 | +2.50 | +7.64 | +0.56 | +1.49 | +16.34 | +2.58 | +14.89 |
| | SoE | +0.94 | +6.57 | +0.51 | −2.06 | +1.64 | +0.64 | +0.13 | +0.44 | −1.98 | +3.83 |
| | CLDL | +2.65 | +1.68 | +1.05 | +0.55 | −7.47 | +2.52 | +9.90 | +7.80 | −7.72 | +15.30 |

Here are some further discussions on the performance. We observe that DVERGE and its iGAT enhancement perform proficiently on both CIFAR10 and CIFAR100, while ADP and its enhancement are less robust on CIFAR100. We attempt to explain this by delving into the algorithm nature of DVERGE and ADP. The ADP design encourages prediction disparities among the base models. As a result, each base model becomes proficient in classifying a subset of classes that the other base models may struggle with. However, a side effect of this is to discourage base models from becoming good at overlapping classes, which may become ineffective when having to handle a larger number of classes. The reasonably good improvement achieved by iGAT for ADP in Table 2 indicates that an addition of global adversarial examples is able to rescue such situation to a certain extent. On the other hand, in addition to encouraging adversarial diversity among the base models, DVERGE also aims at a stable classification so that each example is learned by multiple base models. This potentially makes it suitable for handling both large and small numbers of classes. Moreover, we also observe that the average combiner provides better performance than the max combiner in general.

Table 3: Results of ablation studies based on iGAT$_{ADP}$ using CIFAR-10 under the PGD attack. The results are averaged over five independent runs. The best performance is highlighted in bold.

| | Opposite Distributing | Random Distributing | Hard Distributing | $\beta = 0$ | iGAT$_{ADP}$ |
|---|---|---|---|---|---|
| Natural (%) | 82.45 | 83.05 | 83.51 | 83.45 | **84.96** |
| PGD (%) | 41.31 | 42.60 | 44.21 | 42.32 | **46.25** |

The reason can be that an aggregated prediction from multiple well-trained base classifiers is more statistically stable.

### 6.3 Ablation Studies

The key designs of iGAT include its distributing rule and the regularization term. We perform ablation studies to examine their effectiveness. Firstly, we compare the used soft distributing rule with three alternative distributing rules, including (1) a distributing rule opposite to the proposed, which allocates the adversarial examples to the base models that produce the lowest prediction score, (2) a random distributing rule by replacing Eq. (14) by a uniform distribution, and (3) the hard distributing rule in Eq. (13). Then, we compare with the setting of $\beta = 0$ while keeping the others unchanged. This change removes the proposed regularization term. Results are reported in Table 3 using iGAT$_{ADP}$ with the average combiner, evaluated by CIFAR-10 under the PGD attack. It can be seen that a change or removal of a key design results in obvious performance drop, which verifies the effectiveness of the design.

## 7 Conclusion, Limitation and Future Work

We investigate the challenging and crucial problem of defending against adversarial attacks in the input space of a neural network, with the goal of enhancing ensemble robustness against such attacks while without sacrificing much the natural accuracy. We have provided a formal justification of the advantage of ensemble adversarial defence and proposed an effective algorithmic improvement, bridging the gap between theoretical and practical studies. Specifically, we have proven a decrease in empirical 0-1 loss calculated on data samples challenging to classify, which is constructed to simulate the adversarial attack and defence scenario, under neural network assumptions that are feasible in practice. Also, we have proposed the iGAT approach, applicable to any ensemble adversarial defense technique for improvement. It is supported by (1) a probabilistic distributing rule for selectively allocating global adversarial examples to train base classifiers, and (2) a regularization penalty for addressing vulnerabilities across all base classifiers. We have conducted thorough evaluations and ablation studies using the CIFAR-10 and CIFAR-100 datasets, demonstrating effectiveness of the key designs of iGAT. Satisfactory performance improvements up to $17\%$ have been achieved by iGAT.

However, there is limitation in our work. For instance, our theoretical result is developed for only two base MLPs. We are in progress of broadening the scope of Theorem 4.1 by further relaxing the neural network assumptions, researching model architectures beyond MLPs and beyond the average/max combiners, and more importantly generalizing the theory to more than two base classifiers. Additionally, we are keen to enrich our evaluations using large-scale datasets, e.g., ImageNet. So far, we focus on exploiting curvature information of the loss landscapes to understand adversarial robustness. In the future, it would be interesting to explore richer geometric information to improve the understanding. Despite the research success, a potential negative societal impact of our work is that it may prompt illegal attackers to develop new attack methods once they become aware of the underlying mechanism behind the ensemble cooperation.

### Acknowledgments

We thank the five NeurIPS reviewers for their very insightful and useful comments that help improve the paper draft. We also thank "The University of Manchester - China Scholarship Council Joint Scholarship" for funding Yian's PhD research.

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
