# Supplementary Material for Understanding and Improving Ensemble Adversarial Defense

**Yian Deng**
Department of Computer Science
The University of Manchester
Manchester, UK, M13 9PL
yian.deng@manchester.ac.uk

**Tingting Mu**
Department of Computer Science
The University of Manchester
Manchester, UK, M13 9PL
tingting.mu@manchester.ac.uk

## 1 Studied Ensemble Adversarial Defense Techniques

We briefly explain four ensemble adversarial defense techniques including ADP [5], CLDL [6], DVERGE [7] and SoE [1]. They are used to test the proposed enhancement approach iGAT. In general, an ensemble model contains multiple base models, and the training is conducted by minimizing their classification losses together with a diversity measure. The output of each base model contains the probabilities of an example belonging to the $C$ classes. For an input example $\mathbf{x} \in \mathcal{X}$, we denote its output from the $i$-th base classifier by $\mathbf{h}^i(\mathbf{x}) = \left[h_1^i(\mathbf{x})..., h_C^i(\mathbf{x})\right]$ for $i \in [N]$, where $N$ denotes the base model number.

### 1.1 ADP Defense

ADP employs an ensemble by averaging, i.e., $\mathbf{h}(\mathbf{x}) := \frac{1}{N} \sum_{i=1}^N \mathbf{h}^i(\mathbf{x})$. The base classifiers are trained by minimizing a loss that combines (1) the cross entropy loss of each base classifier, (2) the Shannon entropy of the ensemble prediction for regularization, and (3) a diversity measure to encourage different predictions by the base classifiers. Its formulation is exemplified below using one training example $(\mathbf{x}, y_{\mathbf{x}})$:

$$L_{\text{ADP}}(\mathbf{x}, y_{\mathbf{x}}) = \underbrace{\sum_{i=1}^N \ell_{CE}(\mathbf{h}^i(\mathbf{x}), y_{\mathbf{x}})}_{\text{classification loss}} \underbrace{-\alpha H\left(\mathbf{h}(\mathbf{x})\right)}_{\substack{\text{uncertainty} \\ \text{regularization}}} + \underbrace{\beta \log(D(\mathbf{h}^1(\mathbf{x}), \mathbf{h}^2(\mathbf{x}), \ldots, \mathbf{h}^N(\mathbf{x}), y_{\mathbf{x}}))}_{\text{prediction diversity}},$$

(17)

where $\alpha, \beta \geq 0$ are hyperparameters, the Shannon entropy is $H(\mathbf{p}) = -\sum_{i=1}^C p_i \log(p_i)$, and $D(\mathbf{h}^1, \mathbf{h}^2, \ldots \mathbf{h}^N, y)$ measures the geometric diversity between $N$ different $C$-dimensional probability vectors. To compute the diversity, a normalized $(C-1)$-dimensional vector $\tilde{\mathbf{h}}_{\backslash y}^i$ is firstly obtained by removing from $\mathbf{h}^i$ the element at the position $y \in [C]$, the resulting $\left\{\tilde{\mathbf{h}}_{\backslash y}^i\right\}_{i=1}^N$ are stored as columns of the $(C-1) \times N$ matrix $\tilde{\mathbf{H}}_{\backslash y}$, and then it has $D(\mathbf{h}^1, \mathbf{h}^2, \ldots \mathbf{h}^N, y) = \det\left(\tilde{\mathbf{H}}_{\backslash y}^T \tilde{\mathbf{H}}_{\backslash y}\right)$.

### 1.2 CLDL Defense

CLDL provides an alternative way to formulate the diversity between base classifiers, considering both the base classifier prediction and its loss gradient. Its loss for the training example $(\mathbf{x}, y_{\mathbf{x}})$ is

37th Conference on Neural Information Processing Systems (NeurIPS 2023).

given by

$$L_{\text{CLDL}}(\mathbf{x}, y_{\mathbf{x}}) = \underbrace{\frac{1}{N} \sum_{i=1}^{N} D_{\text{KL}}\left(\mathbf{s}^i(\mathbf{x}) || \mathbf{h}^i(\mathbf{x})\right)}_{\text{classification loss}}$$

$$\underbrace{-\alpha \log \left( \frac{2}{N(N-1)} \sum_{i=1}^{N} \sum_{j=i+1}^{N} e^{\text{JSD}\left(\mathbf{s}_{\backslash}^i(\mathbf{x}) || \mathbf{s}_{\backslash}^j(\mathbf{x})\right)} \right)}_{\text{prediction diversity}}$$

$$+ \underbrace{\frac{2\beta}{N(N-1)} \sum_{i=1}^{N} \sum_{j=i+1}^{N} \cos(\nabla D_{\text{KL}}(\mathbf{s}^i(\mathbf{x}) || \mathbf{h}^i(\mathbf{x})), \nabla D_{\text{KL}}\left(\mathbf{s}^j(\mathbf{x}) || \mathbf{h}^j(\mathbf{x})\right)}_{\text{gradient diversity}}, \quad (18)$$

where $\mathbf{s}^i(\mathbf{x})$ is a soft label vector computed for $(\mathbf{x}, y_{\mathbf{x}})$ by a label smoothing technique called label confusion model [3]. The vector $\mathbf{s}_{\backslash}^i$ is defined as a $(C-1)$-dimensional vector by removing from $\mathbf{s}^i$ its maximal value. The Kullback–Leibler (KL) divergence is used to examine the difference between the soft label vector and the prediction vector, serving as a soft version of the classification loss. The other used divergence measure is Jensen–Shannon divergence (JSD), given as $\text{JSD}(\mathbf{p} || \mathbf{q}) = \frac{1}{2}\left(D_{\text{KL}}(\mathbf{p} || \mathbf{g}) + D_{\text{KL}}(\mathbf{q} || \mathbf{g})\right)$ with $\mathbf{g} = \frac{1}{2}(\mathbf{p} + \mathbf{q})$.

## 1.3 DVERGE Defense

DVERGE proposes a vulnerability diversity to help training the base classifiers with improved adversarial robustness. For training the $i$-th base classifier, it minimizes

$$L_{\text{DVERGE}}^{\text{original}}(\mathbf{x}, y_{\mathbf{x}}) = \underbrace{\ell_{CE}(\mathbf{h}^i(\mathbf{x}), y_{\mathbf{x}})}_{\text{classification loss}} + \underbrace{\alpha \sum_{j \neq i} \mathbb{E}_{(\mathbf{x}_s, y_{\mathbf{x}_s}) \sim D, l \in [L]} \left[ \ell_{CE}\left( \mathbf{h}^i\left( \tilde{\mathbf{x}}\left(\mathbf{h}_{(l)}^j, \mathbf{x}, \mathbf{x}_s\right) \right), y_{\mathbf{x}_s} \right) \right]}_{\text{adversarial vulnerability diversity}},$$
(19)

where $\alpha \geq 0$ is a hyperparameter. Given an input example $\mathbf{x}$, $\tilde{\mathbf{x}}\left(\mathbf{h}_{(l)}^j, \mathbf{x}, \mathbf{x}_s\right)$ computes its distilled non-robust feature vector proposed by Ilyas et al. [4]. This non-robust feature vector is computed with respect to the $l$-th layer of the $j$-th base classifier with its mapping function denoted by $\mathbf{h}_{(l)}^j$ and a randomly sampled natural example $\mathbf{x}_s$, by

$$\tilde{\mathbf{x}}\left(\mathbf{h}_{(l)}^j, \mathbf{x}, \mathbf{x}_s\right) = \arg\min_{\mathbf{z} \in \mathbb{R}^d} \left\| \mathbf{h}_{(l)}^j(\mathbf{z}) - \mathbf{h}_{(l)}^j(\mathbf{x}) \right\|_2^2, \quad (20)$$
$$\text{s.t.} \quad \|\mathbf{z} - \mathbf{x}_s\|_\infty \leq \epsilon.$$

When $\mathbf{x}$ and $\mathbf{x}_s$ belong to different classes, $\tilde{\mathbf{x}}$ can be viewed as an adversarial example that is visually similar to $\mathbf{x}_s$ but is classified by the $j$-th base classifier into the same class as $\mathbf{x}$. This represents a weakness of $\mathbf{h}^j$, and as a correction, the $i$-th base classifier is trained to correctly classify $\tilde{\mathbf{x}}$ into the same class as $\mathbf{x}_s$. But when $\mathbf{x}$ and $\mathbf{x}_s$ come from the same class, $(\tilde{\mathbf{x}}, y_{\mathbf{x}_s})$ is just an example similar to the natural one $(\mathbf{x}_s, y_{\mathbf{x}_s}) \in D$, for which the first and second loss terms play similar roles. Therefore, DVERGE simplifies the above loss in practice, and trains each base classifier by

$$\min_{\mathbf{h}^i} L_{\text{DVERGE}}(\mathbf{x}, y_{\mathbf{x}}) = \mathbb{E}_{(\mathbf{x}_s, y_{\mathbf{x}_s}) \sim D, l \in [L]} \left[ \sum_{j \neq i} \ell_{CE}\left( \mathbf{h}^i\left( \tilde{\mathbf{x}}\left(\mathbf{h}_{(l)}^j, \mathbf{x}, \mathbf{x}_s\right) \right), y_{\mathbf{x}_s} \right) \right]. \quad (21)$$

It removes the classification loss on the natural data.

## 1.4 SoE Defense

SoE proposes a version of classification loss using adversarial examples and a surrogate loss that acts similarly to the vulnerability diversity loss as in DVERGE. For each base classifier $\mathbf{h}^i$, an auxiliary

scalar output head $g^i$ is used to approximate its predicted probability for the true class. Its overall loss exemplified by the training example $(\mathbf{x}, y_{\mathbf{x}})$ is given as

$$L_{\text{SoE}}(\mathbf{x}, y_{\mathbf{x}}) = \underbrace{\sum_{j=1}^{N} \ell_{BCE}\left(h_{y_{\mathbf{x}}}^j\left(\tilde{\mathbf{x}}^i\right), g^j\left(\tilde{\mathbf{x}}^i\right)\right)}_{\text{adversarial classification loss}} - \underbrace{\sigma \ln \sum_{j=1}^{N} \exp\left(\frac{-\ell_{CE}\left(\mathbf{h}^j\left(\tilde{\mathbf{x}}^i\right), y_{\mathbf{x}}\right)}{\sigma}\right)}_{\text{surrogate loss for vulnerability diversity}}, \quad (22)$$

where $\ell_{BCE}$ is the binary cross entropy loss, and $\sigma > 0$ is the weight parameter. Adversarial examples are generated to compute the losses by using the PGD attack. For the $j$-th base classifier, the attack is applied to each $i$-th $(i \neq j)$ base classifer to generate training data, resulting in $\tilde{\mathbf{x}}^i = \phi(\mathbf{h}^i, \mathbf{x}, \text{PGD})$. SoE has two training phases and in the second training phase, rather than using $\tilde{\mathbf{x}}^i$, a different adversarial example is generated by $\tilde{\mathbf{x}} = \phi(\mathbf{h}^k, \mathbf{x}, \text{PGD})$ where $k = \arg\max_{i \in [N]} g^i(\mathbf{x})$, aiming at attacking the best-performing base classifier.

## 2 Proof of Theoretical Results

Given a $C$-class $L$-layer MLP $\mathbf{h} : \mathcal{X} \to [0, 1]^C$ described in Assumption 4.2, we study its cross-entropy loss for one example $(\mathbf{x}, y_{\mathbf{x}})$, i.e., $\ell_{CE}(\mathbf{h}(\mathbf{x}), y_{\mathbf{x}}) = -\log h_{y_{\mathbf{x}}}(\mathbf{x})$, where its partial derivative with respect to the $k$-th element of $\mathbf{x}$ is given by

$$\frac{\partial \ell_{CE}(\mathbf{x})}{\partial x_k} = \sum_{i=1}^{C} (h_i(\mathbf{x}) - \Delta_{i,y_{\mathbf{x}}}) \frac{\partial z_i}{\partial x_k}, \quad (23)$$

where $\Delta_{i,y_{\mathbf{x}}} = \begin{cases} 1, & \text{if } i = y_{\mathbf{x}}, \\ 0, & \text{otherwise.} \end{cases}$ Perturbing the input $\mathbf{x}$ to $\mathbf{x} + \boldsymbol{\delta}$, sometimes we simplify the notation of the perturbed function output, for instance, $\tilde{\ell}(\mathbf{x}) = \ell(\mathbf{x} + \boldsymbol{\delta})$, $\tilde{\mathbf{h}}(\mathbf{x}) = \mathbf{h}(\mathbf{x} + \boldsymbol{\delta})$, $\tilde{\mathbf{z}}(\mathbf{x}) = \mathbf{z}(\mathbf{x} + \boldsymbol{\delta})$ and $\tilde{\sigma}(\mathbf{x}) = \sigma(\mathbf{x} + \boldsymbol{\delta})$.

Our main theorem builds on a supporting Lemma 2.1. In the lemma, we derive an upper bound for the difference between the predictions $\mathbf{h}(\mathbf{x})$ and $\mathbf{h}(\mathbf{z})$ for two examples, computed by an MLP $\mathbf{h} : \mathbb{R}^d \to [0, 1]^C$ satisfying Assumption 4.2. Before proceeding to prove the main theorem, we provide a proof sketch. For each ambiguous pair, we firstly analyse its 0/1 risk under different situations when being classified by a single classifier, and derive its empirical 0/1 risk as $r_1 = 1 - p + \frac{1}{2}p^2$. Then we analyse the 0/1 risk for this pair under different situations when being classified by an ensemble classifier, where both $\max$ and average combiners are considered. We derive the ensemble empirical 0/1 risk as $r_2 = 1 - 3p^2 + 3p^3 - \frac{3}{4}p^4$. Finally, we prove the main result in Eq. (43) by obtaining a sufficient condition for achieving a reduced ensemble risk, i.e., $p > 0.425$ which enables $r_2 \leq r_1$.

### 2.1 Lemma 2.1 and Its Proof

**Lemma 2.1.** *Suppose a $C$-class $L$-layer MLP $\mathbf{h} : \mathbb{R}^d \to [0, 1]^C$ with softmax prediction layer satisfies Assumption 4.2. For any $\mathbf{x}, \mathbf{z} \in \mathbb{R}^d$ and $c = 1, 2 \ldots, C$, the following holds*

$$|h_c(\mathbf{x}) - h_c(\mathbf{z})| \leq \|\mathbf{x} - \mathbf{z}\|_2 B \sqrt{C\left(\tilde{\lambda}^2 - \xi\right)} \quad (24)$$

*for some constant $\xi \leq \tilde{\lambda}^2$, where $\tilde{\lambda}$ and $B$ are constants associated with the MLP family under Assumption 4.2.*

*Proof.* Define the perturbation vector $\boldsymbol{\delta} \in \mathbb{R}^d$ such that $\mathbf{z} = \mathbf{x} + \boldsymbol{\delta}$ and denote its strength by $\epsilon = \|\boldsymbol{\delta}\|_2$, these will be used across the proof. We start from the cross-entropy loss curvature measured by Eq. (2), given as

$$\lambda_{\mathbf{h}}^2(\mathbf{x}, \boldsymbol{\delta}) = \frac{1}{\epsilon^2} \|\nabla \ell_{CE}(\mathbf{h}(\mathbf{x}), y_{\mathbf{x}}) - \ell_{CE}(\mathbf{h}(\mathbf{x} + \boldsymbol{\delta}), y_{\mathbf{x}})\|_2^2 = \frac{1}{\epsilon^2} \sum_k \left(\frac{\partial \ell_{CE}(\mathbf{x})}{\partial x_k} - \frac{\partial \tilde{\ell}_{CE}(\mathbf{x})}{\partial x_k}\right)^2. \quad (25)$$

Below we will expand this curvature expression, where we denote a perturbed function $f(\mathbf{x})$ by using $\tilde{f}(\mathbf{x})$ and $f(\mathbf{x} + \boldsymbol{\delta})$ interchangeably.

By Eq. (23), it has

$$\left| \frac{\partial \ell_{CE}(\mathbf{x})}{\partial x_k} - \frac{\partial \tilde{\ell}_{CE}(\mathbf{x})}{\partial x_k} \right| = \left| \sum_{c=1}^{C} \left( h_c(\mathbf{x}) - \Delta_{i,y_\mathbf{x}} \right) \frac{\partial z_i}{\partial x_k} - \sum_{c=1}^{C} \left( \tilde{h}_c(\mathbf{x}) - \Delta_{i,y_\mathbf{x}} \right) \frac{\partial \tilde{z}_i}{\partial x_k} \right|. \qquad (26)$$

Working with the MLP formulation, it is straightforward to express the quantity $\frac{\partial z_i}{\partial x_k}$ in terms of the derivatives of the activation functions and the neural network weights, as

$$\frac{\partial z_i}{\partial x_k} = \frac{\partial \sum_{s_L} w_{i,s_L}^{(L)} a_{s_L}^{(L-1)}(\mathbf{x})}{\partial x_k} = \sum_{s_L} w_{i,s_L}^{(L)} \frac{\partial a_{s_L}^{(L-1)}(\mathbf{x})}{\partial x_k}. \qquad (27)$$

For the convenience of explanation, we simplify the notation by defining $g_{(L-1),s_L}(\mathbf{x}) = \frac{\partial a_{s_L}^{(L-1)}(\mathbf{x})}{\partial x_k}$, and we have

$$\frac{\partial z_i}{\partial x_k} = \sum_{s_L} w_{i,s_L}^{(L)} g_{(L-1),s_L}(\mathbf{x}), \qquad (28)$$

$$\frac{\partial \tilde{z}_i}{\partial x_k} = \sum_{s_L} w_{i,s_L}^{(L)} \tilde{g}_{(L-1),s_L}(\mathbf{x}). \qquad (29)$$

Applying multivariate Taylor expansion [2], we obtain

$$\tilde{g}_{(L-1),s_L}(\mathbf{x}) = g_{(L-1),s_L}(\mathbf{x}) + \sum_{k=1}^{d} \frac{\partial g_{(L-1),s_L}(\mathbf{x})}{\partial x_k} \delta_k + \sum_{n \geq 2} \left( \sum_{\substack{a_k \in \mathcal{Z}_0, \\ k \in [d], \\ \sum_{k=1}^{d} a_k = n}} C_n^{(a_1,\ldots,a_d)} \delta_1^{a_1} \ldots \delta_d^{a_d} \right), \qquad (30)$$

where $\mathcal{Z}_0$ denotes the set of nonnegative integers, $\delta_k$ is the $k$-th element of the perturbation vector $\boldsymbol{\delta}$, and $C_n^{(a_1,\ldots,a_d)}$ denotes the coefficient of each higher-order term of $\delta_1^{a_1} \ldots \delta_d^{a_d}$. Combining the above equations, we have

$$\sum_k \left( \frac{\partial \ell_{CE}(\mathbf{x})}{\partial x_k} - \frac{\partial \tilde{\ell}_{CE}(\mathbf{x})}{\partial x_k} \right)^2 \qquad (31)$$

$$= \sum_k \left( \sum_{i=1}^{C} \sum_{s_L} \left( h_i(\mathbf{x}) g_{(L-1),s_L}(\mathbf{x}) - \tilde{h}_i(\mathbf{x}) \tilde{g}_{(L-1),s_L}(\mathbf{x}) \right) w_{i,s_L}^{(L)} - \right.$$

$$\left. \sum_{s_L} \left( g_{(L-1),s_L}(\mathbf{x}) - \tilde{g}_{(L-1),s_L}(\mathbf{x}) \right) w_{y_\mathbf{x},s_L}^{(L)} \right)^2$$

$$= \underbrace{\sum_k \left( \sum_{i=1}^{C} \sum_{s_L} \left( h_i(\mathbf{x}) - \tilde{h}_i(\mathbf{x}) \right) g_{(L-1),s_L}(\mathbf{x}) w_{i,s_L}^{(L)} \right)^2}_{T(\mathbf{x})} + \underbrace{\sum_{n \geq 1} \left( \sum_{\substack{a_k \in \mathcal{Z}_0, \\ k \in [d], \\ \sum_{k=1}^{d} a_k = n}} D_n^{(a_1,\ldots,a_d)} \delta_1^{a_1} \ldots \delta_d^{a_d} \right)}_{S(\mathbf{x})},$$

where $D_n^{(a_1,\ldots,a_d)}$ denotes the coefficient of $\delta_1^{a_1} \ldots \delta_d^{a_d}$, computed from the terms like $h_i(\mathbf{x})$, $\tilde{h}_i(\mathbf{x})$, $C_n^{(a_1,\ldots,a_d)}$ and the neural network weights. Define a $C$-dimensional column vector $\mathbf{p}^{(k)}$ with its $i$-th element computed by $p_i^{(k)} = \sum_{s_L} g_{(L-1),s_L}(\mathbf{x}) w_{i,s_L}^{(L)}$ and a matrix $\mathbf{P_h} = \sum_k \mathbf{p}^{(k)} \mathbf{p}^{(k)^T}$, the term $T(\mathbf{x})$ can be rewritten as

$$T(\mathbf{x}) = \sum_k \left( \left( \mathbf{h}(\mathbf{x}) - \tilde{\mathbf{h}}(\mathbf{x}) \right)^T \mathbf{p}_k \right)^2 = \left( \mathbf{h}(\mathbf{x}) - \tilde{\mathbf{h}}(\mathbf{x}) \right)^T \mathbf{P_h} \left( \mathbf{h}(\mathbf{x}) - \tilde{\mathbf{h}}(\mathbf{x}) \right). \qquad (32)$$

The factorization $\mathbf{P_h} = \mathbf{M_h}\mathbf{M_h}^T$ can be obtained by conducting singular value decomposition of $\mathbf{P_h}$. The above new expression of $T(\mathbf{x})$ helps bound the difference between $\mathbf{h}(\mathbf{x})$ and $\tilde{\mathbf{h}}(\mathbf{x})$.

According to the norm definition, we have

$$\|\mathbf{M_h}\|_2 = \max_{\mathbf{q}\in\mathbb{R}^d\neq\mathbf{0}} \frac{\|\mathbf{M_h q}\|_2}{\|\mathbf{q}\|_2} = \max_{\mathbf{q}\in\mathbb{R}^C\neq\mathbf{0}} \frac{\|\mathbf{q}^T\mathbf{M_h}\|_2}{\|\mathbf{q}\|_2}, \tag{33}$$

$$\|\mathbf{M_h^\dagger}\|_2 = \max_{\mathbf{q}\in\mathbb{R}^C\neq\mathbf{0}} \frac{\|\mathbf{M_h^\dagger q}\|_2}{\|\mathbf{q}\|_2} = \max_{\mathbf{q}\in\mathbb{R}^d\neq\mathbf{0}} \frac{\|\mathbf{q}^T\mathbf{M_h^\dagger}\|_2}{\|\mathbf{q}\|_2}. \tag{34}$$

Subsequently, the following holds for any nonzero $\mathbf{q} \in \mathbb{R}^C$ and $\mathbf{p} \in \mathbb{R}^d$

$$\|\mathbf{q}^T\mathbf{M_h}\|_2 \leq \|\mathbf{M_h}\|_2\|\mathbf{q}\|_2, \tag{35}$$

$$\|\mathbf{p}^T\mathbf{M_h^\dagger}\|_2 \leq \|\mathbf{M_h^\dagger}\|_2\|\mathbf{p}\|_2. \tag{36}$$

Letting $\mathbf{q} = \mathbf{h}(\mathbf{x}) - \tilde{\mathbf{h}}(\mathbf{x})$ and using the fact that each element in $\mathbf{h}(\mathbf{x})$ and $\tilde{\mathbf{h}}(\mathbf{x})$ is a probability value less than 1, it has

$$T(\mathbf{x}) = \left\|\left(\mathbf{h}(\mathbf{x}) - \tilde{\mathbf{h}}(\mathbf{x})\right)^T \mathbf{M_h}\right\|_2^2 \leq \|\mathbf{M_h}\|_2^2 \left\|\mathbf{h}(\mathbf{x}) - \tilde{\mathbf{h}}(\mathbf{x})\right\|_2^2 \leq \left(\sup_{\mathbf{h}} \|\mathbf{M_h}\|_2\right)^2 C, \tag{37}$$

which results in the fact that $T(\mathbf{x})$ is upper bounded by Assumption 4.2 where $\|\mathbf{M_h}\|_2 \leq B_0$. Letting $\mathbf{p} = \mathbf{M_h}^T\left(\mathbf{h}(\mathbf{x}) - \tilde{\mathbf{h}}(\mathbf{x})\right)$ and using the Assumption 4.2 where $\left\|\mathbf{M_h^\dagger}\right\|_2 \leq B$, it has

$$\|\mathbf{h}(\mathbf{x}) - \tilde{\mathbf{h}}(\mathbf{x})\|_2 = \left\|\left(\mathbf{h}(\mathbf{x}) - \tilde{\mathbf{h}}(\mathbf{x})\right)^T \mathbf{M_h}\mathbf{M_h^\dagger}\right\|_2$$

$$\leq \left\|\mathbf{M_h^\dagger}\right\|_2 \left\|\left(\mathbf{h}(\mathbf{x}) - \tilde{\mathbf{h}}(\mathbf{x})\right)^T \mathbf{M_h}\right\|_2 \leq B\sqrt{T(\mathbf{x})}. \tag{38}$$

Now we focus on analyzing $T(\mathbf{x})$. Working with Eq. (31) and considering the fact that $\sum_k \left(\frac{\partial\ell_{CE}(\mathbf{x})}{\partial x_k} - \frac{\partial\tilde{\ell}_{CE}(\mathbf{x})}{\partial x_k}\right)^2$ is a positive term and $T(\mathbf{x})$ is upper bounded, $S(\mathbf{x})$ has to be lower bounded. We express this lower bound by $\xi\epsilon^2$ using a constant $\xi$ for the convenience of later derivation, resulting in

$$S(\mathbf{x}) \geq \xi\epsilon^2. \tag{39}$$

Given the perturbation strength $\epsilon^2 = \|\boldsymbol{\delta}\|_2^2$, applying the curvature assumption in Assumption 4.2, i.e., $\lambda_{\mathbf{h}}(\mathbf{x}, \boldsymbol{\delta}) \leq \tilde{\lambda}$, also Eqs. (25), (31) and (39), it has

$$T(\mathbf{x}) + \xi\epsilon^2 \leq \tilde{\lambda}^2\epsilon^2 \Rightarrow T(\mathbf{x}) \leq (\tilde{\lambda}^2 - \xi)\epsilon^2. \tag{40}$$

Incorporating this into Eq. (38), it has

$$\|\mathbf{h}(\mathbf{x}) - \tilde{\mathbf{h}}(\mathbf{x})\|_2 \leq \epsilon B\sqrt{\tilde{\lambda}^2 - \xi}. \tag{41}$$

Applying the inequality of $\sum_{i=1}^m a_i^2 \geq \frac{1}{m}\left(\sum_{i=1}^m a_i\right)^2$, also the fact $\sum_{c=1}^C h_c(\mathbf{x}) = \sum_{c=1}^C \tilde{h}_c(\mathbf{x}) = 1$, the following holds for any class $c \in \{1, 2, \ldots, C\}$:

$$\|\mathbf{h}(\mathbf{x}) - \tilde{\mathbf{h}}(\mathbf{x})\|_2^2 \geq \sum_{j\neq c}\left|h_j(\mathbf{x}) - \tilde{h}_j(\mathbf{x})\right|^2 \geq \frac{1}{C-1}\left(\sum_{j\neq c}\left|h_j(\mathbf{x}) - \tilde{h}_j(\mathbf{x})\right|\right)^2$$

$$\geq \frac{1}{C}\left|\sum_{j\neq c}\left(h_j(\mathbf{x}) - \tilde{h}_j(\mathbf{x})\right)\right|^2 = \frac{1}{C}\left|h_c(\mathbf{x}) - \tilde{h}_c(\mathbf{x})\right|^2. \tag{42}$$

Incorporating Eq. (41) to the above, we have

$$\left|h_c(\mathbf{x}) - \tilde{h}_c(\mathbf{x})\right| \leq \sqrt{C}\|\mathbf{h}(\mathbf{x}) - \tilde{\mathbf{h}}(\mathbf{x})\|_2 \leq \epsilon B\sqrt{C\left(\tilde{\lambda}^2 - \xi\right)}. \tag{43}$$

Inserting back $\mathbf{z} = \mathbf{x} + \boldsymbol{\delta}$ and $\epsilon = \|\boldsymbol{\delta}\|_2$ into Eq. (43), we have

$$|h_c(\mathbf{x}) - h_c(\mathbf{z})| \leq \|\mathbf{x} - \mathbf{z}\|_2 B \sqrt{C\left(\tilde{\lambda}^2 - \xi\right)}. \tag{44}$$

This completes the proof.

$\square$

## 2.2 Proof of Theorem 4.1

**Single Classifier.** We analyse the expected 0/1 risk of a single acceptable classifier $\mathbf{h} \in \mathcal{H}$ for a small dataset $D_2 = \{(\mathbf{x}_i, y_i), (\mathbf{x}_j, y_j)\}$ containing the two examples from the ambiguous pair $a = ((\mathbf{x}_i, y_i), (\mathbf{x}_j, y_j))$. The risk is expressed by

$$\mathbb{E}_{\mathbf{h} \in \mathcal{H}}[\hat{\mathcal{R}}_{0/1}(D_2, \mathbf{h})] = \mathbb{E}_{\mathbf{h} \in \mathcal{H}}\left[\frac{1}{2}\left(1\left[h_{y_i}(\mathbf{x}_i) < \max_{c \neq y_i} h_c(\mathbf{x}_i)\right] + 1\left[h_{y_j}(\mathbf{x}_j) < \max_{d \neq y_j} h_c(\mathbf{x}_j)\right]\right)\right]. \tag{45}$$

We consider three cases.

**Case I**: Suppose the example $(\mathbf{x}_i, y_i)$ is correctly classified, thus, according to Assumption 4.4 for acceptable classifiers, it has $h_{y_i}(\mathbf{x}_i) \geq 0.5 + \frac{1}{J}$. As a result, its prediction score for a wrong class $(c \neq y_i)$ satisfies

$$h_c(\mathbf{x}_i) \leq 1 - h_{y_i}(\mathbf{x}_i) \leq 1 - (0.5 + \frac{1}{J}) = 0.5 - \frac{1}{J} < 0.5 < h_{y_i}(\mathbf{x}_i). \tag{46}$$

Applying Lemma 2.1 for $c = y_i$ and Eq. (10) in Definition 4.3 for ambiguous pair, it has

$$h_{y_i}(\mathbf{x}_i) - h_{y_i}(\mathbf{x}_j) \leq |h_{y_i}(\mathbf{x}_i) - h_{y_i}(\mathbf{x}_j)| \leq \|\mathbf{x}_i - \mathbf{x}_j\|_2 B \sqrt{C\left(\tilde{\lambda}^2 - \xi\right)} \leq \frac{1}{J}. \tag{47}$$

Combining the above with the Case I assumption of $h_{y_i}(\mathbf{x}_i) \geq 0.5 + \frac{1}{J}$, it has

$$h_{y_i}(\mathbf{x}_j) \geq h_{y_i}(\mathbf{x}_i) - \frac{1}{J} \geq (0.5 + \frac{1}{J}) - \frac{1}{J} = 0.5, \tag{48}$$

and hence, for any $c \neq y_i$, it has

$$h_c(\mathbf{x}_j) < 1 - h_{y_i}(\mathbf{x}_j) \leq 0.5 \leq h_{y_i}(\mathbf{x}_j), \tag{49}$$

which indicates that the example $(\mathbf{x}_j, y_j)$ is wrongly predicted to class $y_i$ in Case I. Therefore,

$$\hat{\mathcal{R}}_{0/1}^{(I)}(D_2, \mathbf{h}) = \frac{0 + 1}{2} = \frac{1}{2}. \tag{50}$$

**Case II**: Suppose the example $(\mathbf{x}_j, y_j)$ is correctly classified. Following exactly the same derivation as in Case I, this results in the wrong classification of the other example $(\mathbf{x}_i, y_i)$ into class $y_j$. Therefore,

$$\hat{\mathcal{R}}_{0/1}^{(II)}(D_2, \mathbf{h}) = \frac{1 + 0}{2} = \frac{1}{2}. \tag{51}$$

**Case III**: Suppose both examples are misclassified, which simply results in

$$\hat{\mathcal{R}}_{0/1}^{(III)}(D_2, \mathbf{h}) = \frac{1 + 1}{2} = 1. \tag{52}$$

Note that these three cases are mutually exclusive. Use $E_1$, $E_2$ and $E_3$ to represent the three events corresponding to Case I, Case II and Case III, respectively. Letting $p$ denote the probability of correctly classifying an example by an acceptable classifier, it is straightforward to obtain $p(E_3) = (1 - p)^2$, while $p(E_1) = p(E_2) = \frac{1}{2}\left(1 - (1 - p)^2\right) = p - \frac{1}{2}p^2$. Therefore, it has

$$\mathbb{E}_{\mathbf{h} \in \mathcal{H}}\left[\hat{\mathcal{R}}_{0/1}(D_2, \mathbf{h})\right] \tag{53}$$

$$= \hat{\mathcal{R}}_{0/1}^{(I)}(D_2, \mathbf{h})p(E_1) + \hat{\mathcal{R}}_{0/1}^{(II)}(D_2, \mathbf{h})p(E_2) + \hat{\mathcal{R}}_{0/1}^{(III)}(D_2, \mathbf{h})p(E_3),$$

$$= \frac{1}{2}p(E_1) + \frac{1}{2}p(E_2) + p(E_3) = p - \frac{1}{2}p^2 + (1 - p)^2 = 1 - p + \frac{1}{2}p^2.$$

**Ensemble Classifier.** We next analyse using $D_2$ the expected 0/1 risk of an ensemble of two acceptable base classifiers ($\mathbf{h}^0, \mathbf{h}^1 \in \mathcal{H}$) with a *max* or average combiner, in five cases.

**Case I**: Suppose the example $(\mathbf{x}_i, y_i)$ is correctly classified by both base classifiers. According to Assumption 4.4 for acceptable classifiers, it has $h_{y_i}^0(\mathbf{x}_i) \geq 0.5 + \frac{1}{J}$ and $h_{y_i}^1(\mathbf{x}_i) \geq 0.5 + \frac{1}{J}$. Following exactly the same derivation as in the earlier Case I analysis for a single classifier, i.e., Eqs. (46) and (49), the following holds for any $c \neq y_i$, as

$$h_c^0(\mathbf{x}_i) < h_{y_i}^0(\mathbf{x}_i), \; h_c^0(\mathbf{x}_j) < h_{y_i}^0(\mathbf{x}_j), \tag{54}$$

$$h_c^1(\mathbf{x}_i) < h_{y_i}^1(\mathbf{x}_i), \; h_c^1(\mathbf{x}_j) < h_{y_i}^1(\mathbf{x}_j). \tag{55}$$

As a result, for any $c \neq y_i$, the ensemble prediction satisfies the following

$$h_{e,y_i}^{(0,1)}(\mathbf{x}_i) = \max\left(h_{y_i}^0(\mathbf{x}_i), h_{y_i}^1(\mathbf{x}_i)\right) > \max(h_c^0(\mathbf{x}_i), h_c^1(\mathbf{x}_i)) = h_{e,c}^{(0,1)}(\mathbf{x}_i), \tag{56}$$

$$h_{e,y_i}^{(0,1)}(\mathbf{x}_i) = \frac{1}{2}\left(h_{y_i}^0(\mathbf{x}_i) + h_{y_i}^1(\mathbf{x}_i)\right) > \frac{1}{2}(h_c^0(\mathbf{x}_i) + h_c^1(\mathbf{x}_i)) = h_{e,c}^{(0,1)}(\mathbf{x}_i), \tag{57}$$

each corresponding to the $\max$ and average combiners, respectively. This indicates a correct ensemble classification of $(\mathbf{x}_i, y_i)$. Also, it satisfies

$$h_{e,y_j}^{(0,1)}(\mathbf{x}_j) = \max\left(h_{y_j}^0(\mathbf{x}_j), h_{y_j}^1(\mathbf{x}_j)\right) < \max(h_{y_i}^0(\mathbf{x}_j), h_{y_i}^1(\mathbf{x}_j)) = h_{e,y_i}^{(0,1)}(\mathbf{x}_j), \tag{58}$$

$$h_{e,y_j}^{(0,1)}(\mathbf{x}_j) = \frac{1}{2}\left(h_{y_j}^0(\mathbf{x}_j) + h_{y_j}^1(\mathbf{x}_j)\right) < \frac{1}{2}(h_{y_i}^0(\mathbf{x}_j) + h_{y_i}^1(\mathbf{x}_j)) = h_{e,y_i}^{(0,1)}(\mathbf{x}_j), \tag{59}$$

when using the $\max$ and average combiners, respectively. This indicates a wrong classification of $(\mathbf{x}_j, y_j)$. Finally, for Case I, we have

$$\hat{\mathcal{R}}_{0/1}^{(\text{I})}\left(D_2, \mathbf{h}_e^{(0,1)}\right) = \frac{1}{2}(0+1) = \frac{1}{2}, \tag{60}$$

**Case II**: Suppose the example $(\mathbf{x}_j, y_j)$ is correctly classified by both base classifiers. By following exactly the same derivation as in Case I as above, the ensemble correctly classifies $(\mathbf{x}_j, y_j)$, while wrongly classifies $(\mathbf{x}_i, y_i)$. As a result, it has

$$\hat{\mathcal{R}}_{0/1}^{(\text{II})}\left(D_2, \mathbf{h}_e^{(0,1)}\right) = \frac{1}{2}(1+0) = \frac{1}{2}. \tag{61}$$

**Case III**: Suppose the example $(\mathbf{x}_i, y_i)$ is correctly classified by $\mathbf{h}^0$, while the other example $(\mathbf{x}_j, y_j)$ is correctly classified by $\mathbf{h}^1$, i.e., $h_{y_i}^0(\mathbf{x}_i) \geq 0.5 + \frac{1}{J}$ and $h_{y_j}^1(\mathbf{x}_j) \geq 0.5 + \frac{1}{J}$ according to Assumption 4.4. Following a similar analysis as in Case I for a single classifier, we know that $\mathbf{h}^0$ consequently misclassifies $(\mathbf{x}_j, y_j)$ into $y_i$, while $\mathbf{h}^1$ misclassifies $(\mathbf{x}_i, y_i)$ into $y_j$. Also, by Assumption 4.4, it is assumed that the misclassification happens with a less score than $0.5 + \frac{1}{J}$, thus, $h_{y_i}^0(\mathbf{x}_j) \leq 0.5 + \frac{1}{J}$ and $h_{y_j}^1(\mathbf{x}_i) \leq 0.5 + \frac{1}{J}$. Combining all these, for any $c \neq y_i$ and $d \neq y_j$, we have

$$h_d^1(\mathbf{x}_i) < 0.5 \leq h_{y_j}^1(\mathbf{x}_i) \leq 0.5 + \frac{1}{J} \leq h_{y_i}^0(\mathbf{x}_i), \tag{62}$$

$$h_c^0(\mathbf{x}_j) < 0.5 \leq h_{y_i}^0(\mathbf{x}_j) \leq 0.5 + \frac{1}{J} \leq h_{y_j}^1(\mathbf{x}_j), \tag{63}$$

and according to the second condition in Assumption 4.4, it has

$$h_c^0(\mathbf{x}_i) \leq \frac{1 - h_{y_i}^0(\mathbf{x}_i)}{C-1} \leq h_{y_i}^0(\mathbf{x}_i), \tag{64}$$

$$h_d^1(\mathbf{x}_j) \leq \frac{1 - h_{y_j}^1(\mathbf{x}_j)}{C-1} \leq h_{y_j}^1(\mathbf{x}_j). \tag{65}$$

Subsequently, the ensemble prediction by a $\max$ combiner satisfies

$$h_{e,y_i}^{(0,1)}(\mathbf{x}_i) = \max\left(h_{y_i}^0(\mathbf{x}_i), h_{y_i}^1(\mathbf{x}_i)\right) = h_{y_i}^0(\mathbf{x}_i) > \max(h_c^0(\mathbf{x}_i), h_c^1(\mathbf{x}_i)) = h_{e,c}^{(0,1)}(\mathbf{x}_i), \tag{66}$$

$$h_{e,y_j}^{(0,1)}(\mathbf{x}_j) = \max\left(h_{y_j}^0(\mathbf{x}_j), h_{y_j}^1(\mathbf{x}_j)\right) = h_{y_j}^1(\mathbf{x}_j) > \max(h_d^0(\mathbf{x}_j), h_d^1(\mathbf{x}_j)) = h_{e,d}^{(0,1)}(\mathbf{x}_j), \tag{67}$$

which indicates a correct classification of both examples.

Now we consider the slightly more complex situation of ensemble by averaging. According to the previous analysis, we know that $\mathbf{x}_i$ is classified by $\mathbf{h}^1$ to $y_j$, and $\mathbf{x}_i$ is classified by $\mathbf{h}^0$ to $y_i$. Applying the second condition in Assumption 4.4, we analyse the quantity $1 - h^1_{y_j}(\mathbf{x}_i) - h^1_{y_i}(\mathbf{x}_i)$ as

$$1 - h^1_{y_j}(\mathbf{x}_i) - h^1_{y_i}(\mathbf{x}_i) = \sum_{c \neq y_i, y_j} h^1_c(\mathbf{x}_i) \leq (C-2)\frac{1 - h^1_{y_j}(\mathbf{x}_i)}{C-1} = 1 - h^1_{y_j}(\mathbf{x}_i) - \left(\frac{1 - h^1_{y_j}(\mathbf{x}_i)}{C-1}\right), \quad (68)$$

resulting in

$$h^1_{y_i}(\mathbf{x}_i) \geq \frac{1 - h^1_{y_j}(\mathbf{x}_i)}{C-1}. \quad (69)$$

Combining Eq. (62), Eq. (64) and Eq. (69), it has

$$h^1_{y_i}(\mathbf{x}_i) \geq \frac{1 - h^1_{y_j}(\mathbf{x}_i)}{C-1} > \frac{1 - h^0_{y_i}(\mathbf{x}_i)}{C-1} \geq h^0_c(\mathbf{x}_i). \quad (70)$$

On the other hand, from Eq. (62), one can obtain

$$h^0_{y_i}(\mathbf{x}_i) \geq h^1_c(\mathbf{x}_i). \quad (71)$$

As a result, the ensemble prediction by an average combiner satisfies

$$h^{(0,1)}_{e,y_i}(\mathbf{x}_i) = \frac{1}{2}\left(h^0_{y_i}(\mathbf{x}_i) + h^1_{y_i}(\mathbf{x}_i)\right) > \frac{1}{2}\left(h^0_c(\mathbf{x}_i) + h^1_c(\mathbf{x}_i)\right) = h^{(0,1)}_{e,c}(\mathbf{x}_i), \quad (72)$$

for any $c \neq y_i$. Following the same way of deriving Eqs. (70) and (71), but for $\mathbf{x}_j$, we can obtain another two inequalities $h^1_{y_j}(\mathbf{x}_j) \geq h^0_d(\mathbf{x}_j)$ and $h^0_{y_j}(\mathbf{x}_j) \geq h^1_d(\mathbf{x}_j)$, for any $d \neq y_j$, and subsequently,

$$h^{(0,1)}_{e,y_j}(\mathbf{x}_j) = \frac{1}{2}\left(h^0_{y_j}(\mathbf{x}_j) + h^1_{y_j}(\mathbf{x}_j)\right) > \frac{1}{2}\left(h^0_d(\mathbf{x}_j) + h^1_d(\mathbf{x}_j)\right) = h^{(0,1)}_{e,d}(\mathbf{x}_j). \quad (73)$$

Putting together Eqs. (72) and (73), a correct ensemble classification is achieved for both examples. Finally, we conclude the following result

$$\hat{\mathcal{R}}^{(\text{III})}_{0/1}\left(D_2, \mathbf{h}^{(0,1)}_e\right) = 0, \quad (74)$$

which is applicable to both the max and average combiners.

**Case IV**: Suppose the example $(\mathbf{x}_i, y_i)$ is correctly classified by $\mathbf{h}^1$ while the other example $(\mathbf{x}_j, y_j)$ is correctly classified by $\mathbf{h}^0$. This is essentially the same situation as in Case III, and the same result $\hat{\mathcal{R}}^{(\text{IV})}_{0/1}\left(D_2, \mathbf{h}^{(0,1)}_e\right) = 0$ is obtained.

**Case V**: This case represents all the remaining situations, where, for instance, the example $(\mathbf{x}_i, y_i)$ and/or $(\mathbf{x}_i, y_i)$ is misclassified by both base classifiers. Here, we do not have sufficient information to analyse the error in detail, and also it is not necessary to do so for our purpose. So we just simply leave it as $\hat{\mathcal{R}}^{(\text{V})}_{0/1}\left(D_2, \mathbf{h}^{(0,1)}_e\right) \leq 1$.

These five cases are mutually exclusive, and we use $\{H_i\}^5_{i=1}$ to denote them accordingly. The first four cases represent the same situation that each example is correctly classified by a single base classifier, therefore $p(H_1) = p(H_2) = p(H_3) = p(H_4) = p(E_1)p(E_2) = \left(p - \frac{1}{2}p^2\right)^2$, while $p(H_5) = 1 - \sum^4_{i=1} p(H_i) = 1 - 4\left(p - \frac{1}{2}p^2\right)^2 = 1 - (2p - p^2)^2$. Incorporating the result of $\hat{\mathcal{R}}_{0/1}\left(D_2, \mathbf{h}^{(0,1)}_e\right)$ regarding to the five cases, we have

$$\mathbb{E}_{\mathbf{h}^0, \mathbf{h}^1 \in \mathcal{H}}\left[\hat{\mathcal{R}}_{0/1}\left(D_2, \mathbf{h}^{(0,1)}_e\right)\right] \quad (75)$$

$$\leq \frac{1}{2}p(H_1) + \frac{1}{2}p(H_2) + 0(p(H_3) + p(H_4)) + p(H_5)$$

$$= \left(p - \frac{1}{2}p^2\right)^2 + 1 - (2p - p^2)^2 = 1 - 3p^2 + 3p^3 - \frac{3}{4}p^4.$$

**Risk Comparison.** We examine the sufficient condition for achieving a reduced ensemble loss for this dataset $D_2$, i.e.,

$$\mathbb{E}_{\mathbf{h}^0,\mathbf{h}^1 \in \mathcal{H}} \left[ \hat{\mathcal{R}}_{0/1} \left( D_2, \mathbf{h}_e^{(0,1)} \right) \right] < \mathbb{E}_{\mathbf{h} \in \mathcal{H}} \left[ \hat{\mathcal{R}}_{0/1} \left( D_2, \mathbf{h} \right) \right]. \tag{76}$$

Incorporating Eqs. (53) and (75), this requires to solve the following polynomial inequality, as

$$1 - 3p^2 + 3p^3 - \frac{3}{4}p^4 < 1 - p + \frac{1}{2}p^2, \tag{77}$$

for which $p > 0.425$ provides a solution. Applying the expectation $\mathbb{E}_{a \sim A(D)}$ over the data samples, where the ambiguous pair $a$ is equivalent to $D_2$, Eq. (5) from the theorem is obtained. This completes the proof.

# 3 A Toy Example for Theorem 4.1

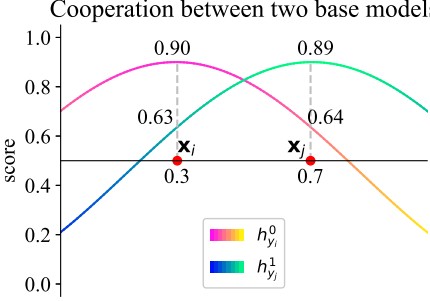

Figure 1: Illustration for Theorem 4.1.

Guided by Theorem 4.1, we aim to verify the difference between the single-branch and ensemble mechanisms using 1-dimensional 2-class data. Suppose $D = \{(\mathbf{x}_i = 0.3, y_i = 0), (\mathbf{x}_j = 0.7, y_j = 1)\}$, we use it to construct an ambiguous pair $a = ((\mathbf{x}_i, y_i), (\mathbf{x}_j, y_j))$ as presented in Fig. 1. We select two base models $\mathbf{h}^0, \mathbf{h}^1 \in \mathcal{H}$ such that $\mathbf{h}^0$ classifies $\mathbf{x}_i$ well and $\mathbf{h}^1$ classifies $\mathbf{x}_j$ well. W.l.o.g, let $\mathbf{h} = \mathbf{h}^0$. For the classifiers $\mathbf{h}$, $\mathbf{h}^0$ and $\mathbf{h}^1$, we analyze the 0-1 loss defined in Eq. (4). Then, we have

$$\hat{\mathcal{R}}_{0/1}(a, \max(\mathbf{h}^0, \mathbf{h}^1)) = \frac{1}{2} \big( 1 \left[ \max(0.9, 0.37) < \max(0.1, 0.63) \right]$$
$$+ 1 \left[ \max(0.89, 0.36) < \max(0.11, 0.64) \right] \big) = 0,$$
$$\hat{\mathcal{R}}_{0/1}(a, \left( \mathbf{h}^0 + \mathbf{h}^1 \right)/2) = \frac{1}{2} \big( 1 \left[ (0.9 + 0.37)/2 < (0.1, 0.63)/2 \right]$$
$$+ 1 \left[ (0.89 + 0.36)/2 < (0.11 + 0.64)/2 \right] \big) = 0,$$
$$\hat{\mathcal{R}}_{0/1}(a, \mathbf{h}) = \frac{1}{2} \big( 1 \left[ 0.9 < 0.63 \right] + 1 \left[ 0.36 < 0.64 \right] \big) = 0.5.$$

Hence, it has $\hat{\mathcal{R}}_{0/1}(a, \max(\mathbf{h}^0, \mathbf{h}^1)) < \hat{\mathcal{R}}_{0/1}(a, \mathbf{h})$ and $\hat{\mathcal{R}}_{0/1}(a, \left( \mathbf{h}^0 + \mathbf{h}^1 \right)/2) < \hat{\mathcal{R}}_{0/1}(a, \mathbf{h})$, which matches the resulting inequality in Theorem 4.1.

# 4 Additional Experiments and Results

**Extra Black-box Attacks:** We conduct more experiments to test the effectiveness of iGAT, by evaluating against another two time-efficient and commonly used black-box attacks, using the CIFAR-10 dataset. Results are reported in Table 1. It can be seen that, in most cases, a robustness improvement has been achieved by the enhanced defence.

Table 1: Results on two additional black-box attacks.

| | Simple Attack (%) | Bandits Attack (%) |
|---|---|---|
| ADP | 75.91 | 59.21 |
| iGAT$_{ADP}$ | **79.43** | **64.55** |
| DVERGE | 79.43 | 63.80 |
| iGAT$_{DVERGE}$ | **79.61** | **64.89** |
| CLDL | 76.82 | 63.80 |
| iGAT$_{CLDL}$ | **78.84** | **65.25** |
| SoE | **76.22** | 66.10 |
| iGAT$_{SoE}$ | 75.18 | **66.50** |

Table 2: Comparison of the ensemble robustness (%) to adversarial attacks of various perturbation strengths, using the AutoAttack on CIFAR-10. The results are averaged over five independent runs.

| | $\epsilon$ | $2/255$ | $4/255$ | $6/255$ | $8/255$ | $10/255$ |
|---|---|---|---|---|---|---|
| CIFAR10 | CLDL | 71.16 | 60.36 | 48.89 | 37.06 | 26.00 |
| | iGAT$_{CLDL}$ | **72.69** | **61.19** | **49.07** | **37.12** | 25.96 |
| | DVERGE | 76.01 | 64.80 | 51.92 | 39.22 | 27.72 |
| | iGAT$_{DVERGE}$ | .**76.19** | **65.14** | **52.52** | **39.48** | **28.59** |
| | ADP | 71.93 | 59.53 | 47.27 | 35.52 | 25.01 |
| | iGAT$_{ADP}$ | **76.02** | **64.76** | **52.44** | **40.38** | **29.46** |
| CIFAR100 | SoE | 46.55 | 33.89 | 23.77 | 15.92 | 10.49 |
| | iGAT$_{SoE}$ | 45.72 | 33.18 | 23.28 | **16.09** | **10.52** |
| | DVERGE | 48.87 | 35.81 | 25.35 | 17.26 | 11.18 |
| | iGAT$_{DVERGE}$ | **49.43** | **37.11** | **26.78** | **18.60** | **12.13** |
| | ADP | 45.67 | 33.90 | 24.42 | 17.36 | 12.27 |
| | iGAT$_{ADP}$ | **46.33** | **34.33** | **24.85** | **17.86** | **12.53** |

**Varying Perturbation Strengths:** In addition to the perturbation strength $\epsilon = 8/255$ tested in the main experiment, we compare the defense techniques under AutoAttack with different settings of perturbation strength. Table 2 reports the resulting classification accuraccies, demonstrating that the proposed iGAT is able to improve the adversarial robustness of the studied defense techniques in most cases.

**Comparison Against Single Classifiers:** To observe how an ensemble classifier performs with specialized ensemble adversarial training, we compare iGAT$_{ADP}$ based on the average combiner against a single-branch classifier. This classifier uses the ResNet-18 architecture, and is trained using only the standard adversarial training without any diversity or regularization driven treatment. Table 3 reports the results. It can be seen that the specialized ensemble adversarial training technique can significantly improve both the natural accuracy and adversarial robustness.

**Experiments Driven by Assumption 4.4:** To approximate empirically the probability $p$ that a trained base classifier can correctly classify a challenging example, we generate a set of globally adversarial examples $\tilde{\mathbf{X}}$ by attacking the ensemble $\mathbf{h}$ (average combiner) using the PGD and then estimate $p$ on this dataset by $p = \mathbb{E}_{i \in [N], (\mathbf{x}, y_\mathbf{x}) \sim (\tilde{\mathbf{X}}, \mathbf{y})} 1[h_{y_\mathbf{x}}^i(\mathbf{x}) > \max_{c \neq y_\mathbf{x}} h_c^i(\mathbf{x})]$. From Table 4, we can see that all the enhanced ensembles contain base models with a higher probability for correct classifications.

We then examine the distributions of predicted scores by base models when classifying correctly the globally adversarial data generated in the same as in Table 4. It can be seen that the case exists, where a base model correctly classifies a challenging example with a sufficiently large predicted score.

Table 3: Comparison between iGAT$_{\text{ADP}}$ (average combiner) and a baseline single classifier, evaluated using CIFAR-10 data and the PGD attack ($\epsilon = 8/255$). The results are averaged over five independent runs.

| | Natural (%) | PGD (%) | Model size |
|---|---|---|---|
| Single Classifier | 81.23 | 38.33 | 43M |
| iGAT$_{\text{ADP}}$ | **84.95** | **46.25** | 9M |

Table 4: Probabilities of base models classifying correctly adversarial examples from the CIFAR-10.

| | ADP | iGAT$_{\text{ADP}}$ | DVERGE | iGAT$_{\text{DVERGE}}$ | CLDL | iGAT$_{\text{CLDL}}$ |
|---|---|---|---|---|---|---|
| $p$ | 41.92% | 45.98% | 46.25% | 47.82% | 50.37% | 51.02% |

Table 5: Distributions of predicted scores by base models correctly classifying adversarial examples from the CIFAR-10.

| Interval | <0.5 | 0.5-0.6 | 0.6-0.7 | 0.7-0.8 | 0.8-0.9 | 0.9-1.0 |
|---|---|---|---|---|---|---|
| iGAT$_{\text{ADP}}$ | 43.55% | 13.30% | 11.15% | 10.20% | 10.62% | 11.19% |
| iGAT$_{\text{DVERGE}}$ | 20.07% | 13.15% | 12.20% | 12.46% | 14.44% | 27.69% |
| iGAT$_{\text{CLDL}}$ | 49.50% | 14.12% | 12.26% | 13.53% | 9.77% | 0.81% |

Table 6: Expectations of the maximum predicted scores on incorrect classes among base models when tested on adversarial examples from the CIFAR-10.

| ADP | iGAT$_{\text{ADP}}$ | DVERGE | iGAT$_{\text{DVERGE}}$ | CLDL | iGAT$_{\text{CLDL}}$ |
|---|---|---|---|---|---|
| 0.390 | 0.323 | 0.476 | 0.396 | 0.320 | 0.281 |

Next, we compute the quantity, i.e., the largest incorrectly predicted score $\mathbb{E}_{i\in[N],(\mathbf{x},\mathbf{y}_{\mathbf{x}})\sim(\tilde{\mathbf{X}},\mathbf{y})}\max_{c\neq y_{\mathbf{x}}}h_c^i(\mathbf{x})$, to indirectly estimate whether the small-incorrect-prediction condition, i.e., $f_c(\mathbf{x}) \leq \frac{1-f_{\hat{y}}(\mathbf{x})}{C-1}$ in Assumption 4.4, can be satisfied better after enhancement. Note that $y_i \neq \hat{y}_i$ indicates the incorrect classification while $y_i = \hat{y}_i$ indicates the opposite, both of which are uniformly measured by the defined quantity. This quantity, which is expected to be small, can also be used to evaluate the effect of the proposed regularization term in Eq. (15) on the training. Table 6 shows that the largest wrongly predicted scores by the base models have significantly dropped for all the enhanced ensemble models.

Note that small values of $h_{c\neq y_{\mathbf{x}}}^i(\mathbf{x})$ is equivalent to the high values of $h_{y_{\mathbf{x}}}^i(\mathbf{x})$, and in the theorem, when $\hat{y} \neq y_{\mathbf{x}}$, $h_{y_{\mathbf{x}}}^i(\mathbf{x}) \geq \frac{1-h_{\hat{y}}^i(\mathbf{x})}{C-1}$ is the actual condition expected to be satisfied. Therefore, to examine the second item (the case of misclassification) in Assumption 4.4, we measure the probability $\mathbb{E}_{i\in[N],(\mathbf{x},y_{\mathbf{x}})\sim(\tilde{\mathbf{X}},\mathbf{y})}\mathbb{1}\left[h_{y_{\mathbf{x}}}^i(\mathbf{x}) \geq \frac{1-h_{\hat{y}}^i(\mathbf{x})}{C-1}\right]$ instead. Table 7 shows that after enhancement, the probability of satisfying the condition increases.

As shown in Figure 1, as long as the peaks of two curves are above the line $x = 0.5$ and at similar heights (in which case, are 0.89 and 0.90), whether their height are changed slightly to a higher or lower position will not increase the 0-1 loss. Elevating the low predicted scores to the same level as the high scores serves the crucial factor in fulfilling the cooperative function. Hence, we choose to examine the effect of our distributing rule by checking whether the predicted scores by the best-performing base models on incorrectly classified examples have been increased after enhancement, using the quantity $\mathbb{E}_{(\mathbf{x},y_{\mathbf{x}})\sim(\tilde{\mathbf{X}},\mathbf{y}),\hat{y}_{\mathbf{h}}(\mathbf{x})\neq y_{\mathbf{x}}}[\max_{i\in[N]}h_{y_{\mathbf{x}}}^i(\mathbf{x})]$. It can be seen from Table 8 that base models were kept improved on the examples they are already good at classifying.

**Time Efficiency of iGAT:** (1) On distributing rule: We expect the distributing rule to reduce the training data size to $\frac{1}{N}$ for training each base classifier, where $N$ is the number of base classifiers, and therefore to improve the training time. We add an experiment by comparing the training time on

Table 7: Probabilities of $h^i_{y_\mathbf{x}}(\mathbf{x}) \geq \frac{1-h^i_{\hat{y}}(\mathbf{x})}{C-1}$ for $y_\mathbf{x} \neq \hat{y}$ when tested on adversarial examples from the CIFAR-10.

| ADP | iGAT$_{\text{ADP}}$ | DVERGE | iGAT$_{\text{DVERGE}}$ | CLDL | iGAT$_{\text{CLDL}}$ |
|---|---|---|---|---|---|
| 68.74% | 73.12% | 78.99% | 80.19% | 78.39% | 80.87% |

Table 8: Predicted scores on incorrectly classified adversarial examples by the best-performing base model using the CIFAR-10.

| ADP | iGAT$_{\text{ADP}}$ | DVERGE | iGAT$_{\text{DVERGE}}$ | CLDL | iGAT$_{\text{CLDL}}$ |
|---|---|---|---|---|---|
| 0.264 | 0.291 | 0.231 | 0.240 | 0.235 | 0.241 |

$N = 1000$ training samples required by a full version of iGAT$_{\text{ADP}}$ and that by a modified version with this distributing rule removed. CIFAR-10 data is used for Evaluation. The observed time for iGAT$_{\text{ADP}}$ without the distributing design is $5.63$ seconds, while with the distributing design is $5.42$ seconds, indicating a slightly reduced training time. (2) On overall training: We illustrate the training epochs between the ADP defense and its enhancement iGAT$_{\text{ADP}}$. ADP necessitates 691 epochs for ADP, whereas iGAT$_{\text{ADP}}$ only requires 163 epochs. Based on these, we can conclude that iGAT$_{\text{ADP}}$ trains faster than ADP.

**Observation of Curvature:** We investigated empirically the value of the network curvature $\tilde{\lambda}$ using neural networks trained by the ADP defense techniques, and recorded a $\tilde{\lambda}$ value around $0.06$. The smaller value of $\tilde{\lambda}$ indicates a looser upper bound in Eq. (10). According to our Definition 4.3, a looser upper bound allows to define an ambiguous pair containing two intra-class examples that are less close to each other, thus less challenging to classify.