# OpenReview forum: "Understanding and Improving Ensemble Adversarial Defense"
_NeurIPS.cc/2023/Conference — NeurIPS 2023 poster_

### Official Review · Reviewer_d7H2 · 2023-07-03

**Soundness:** 2 fair
**Presentation:** 3 good
**Contribution:** 2 fair
**Rating:** 5
**Confidence:** 4

**Summary:**

This paper first propose to understand ensemble adversarial defense from a theoretical perspective. Then the authors develop iGAT, including a distributing strategy is to encourage each base classifier to keep improving over regions that they are relatively good at classifying. This leads to a hard distributing rule that assigning in a deterministic way the example to the base classifier that returns the highest predicted probability on its ground truth class, and a soft distributing rule based on a ranking-based score. Besides, a  regularization term is introduced to address the severest weakness,  by minimizing the probability score of the most incorrectly predicted class by the most erroneous base classifier. Empirical evaluations are done on enhancing previous ensemble defenses.

**Strengths:**

This paper provides a theoretical analysis on why an ensemble of adversarially trained classifiers is more robust than single ones and develop a new error theory dedicated to understanding ensemble adversarial defense. This theoretical analysis provides some new insights on the effectiveness of ensemble defense.

Inspired from the theory, the authors propose iGAT, including probabilistic distributing rule and a regularization term for rescuing the severest weaknesses of the base classifiers. Empirically, iGAT is shown to consistently enhance previous ensemble defense strategies such as ADP, CLDL, DVERGE and SoE.

**Weaknesses:**

I have some concerns:

- The designs of distributing rule in Section 5.1 and regularization in Section 5.2 are somewhat heuristic. For example, it is not well-justified on why ranking-based probability in Equation (10) is preferred compared to other soft distribution.

- Does the hyperparameters \alpha and \beta have to be the same in Equation (7) and Equation (12)? If yes, please elaborate on why they have to be the same? If not, then there are too many hyperparameters (e.g., four) to tune.

- In Table 1 (CIFAR-10), the empirical improvements of iGAT_DVERGE vs. DVERGE and iGAT_CLDL vs. CLDL are quite marginal under AA.

- Besides, I observe that in Table 1, iGAT is combined with CLDL, DVERGE, and ADP on CIFAR-10; while iGAT is combined with SoE, ADP, DVERGE on CIFAR-100. Is there any special reason for this experimental setting?

**Questions:**

My questions are listed in Weaknesses.

**Limitations:**

Limitations are not discussed in the paper. As to the potential negative societal impact, there is a sentence in Conclusion.

---

> ### Author Rebuttal · Authors · 2023-08-09
>
> We thank you for the insightful comments, and address them below.
>
> # Weaknesses
>
> * **On Weakness 1.** Both our ranking and regularization designs are motivated by Assumption 4.4 (``Acceptable Classifier'') which identifies sufficient conditions that should be satisfied by a base classifier in order to guarantee a reduced 0/1 risk after ensemble. Part two of the assumption expects the classifier to have small prediction scores on wrong classes. An effective way to achieve this is by the proposed regularization. Part one of the assumption expects the base classifiers to perform well on ambiguous pairs. To promote this, we thus distribute each training example to its best-performing base classifier. The most straightforward way to find the best base classifier is through ranking their prediction scores, and Eq. (10) is an effective way of using such ranking information while being robust to prediction errors.  There can be other ways to design the distributing rules with similar goals, but we have found our design sufficient through empirical evaluations. Our ablation study (2) in Section 6.3 supports the validity of Eq. (10) design.
>
> * **On Weakness 2.** Indeed there are four hyper-parameters in total, and we should have used different notations for the two in either Eq. (6) or Eq. (12), instead of $\alpha$ and $\beta$. However,  the two hyper-parameters in Eq. (6) do not need to be tuned, for which any default or recommended setting fixed in the adopted adversarial ensemble baseline can be used. We tune only the two iGAT hyper-parameters in Eq. (12).
>
> * **On Weakness 3.**  Although the improvement is marginal, it is still positive.  At this stage, it is hard to form a theory to predict for which method iGAT will perform better, but we will gather more insight on this as our research progresses further in the future.
>
> * **On Weakness 4.** We compared using the top 3 performing methods for each dataset. Encouraged by this comment and also another comment from another reviewer, we have conducted experiments to include comparisons for all the four baseline methods using the average combiner. Please see the added results in our reply to Question 2 of Reviewer cK43 (Reviewer 2).

---

> > ### Comment · Reviewer_d7H2 · 2023-08-13
> >
> > I thank the authors for providing responses to my questions. However, my concerns are not fully addressed:
> >
> > - About Weakness 1, there are many designs of distribution rule and regularization term that can satisfy the properties in Assumption 4.4, and my question was: why the ones proposed in this paper (e.g., ranking-based probability in Eq. (10)) are preferred compared to other alternatives? For example, it would be great if the authors could propose a reasonable metric and demonstrate that the ranking-based probability achieves optimality under this metric.
> >
> > - About Weakness 2 and 3, four hyperparameters seem too many for me; even though some of them can be set by default, it is not guaranteed that the model performance is unaffected by them on other datasets or scenarios. Furthermore, the marginal improvements are also not convinced enough.
> >
> > So overall, my final rating is still a borderline accept.

---

### Official Review · Reviewer_XCW5 · 2023-07-05

**Soundness:** 3 good
**Presentation:** 3 good
**Contribution:** 3 good
**Rating:** 5
**Confidence:** 2

**Summary:**

This paper derives provable error reduction when shifting from employing one neural network to an ensemble of two neural networks, which provides new insights into ensemble adversarial defenses. Based on the theoretical findings, the authors propose iGAT to narrow the subspace where all base classifiers misclassify inputs.

**Strengths:**

1. This paper proposes a new understanding of ensemble adversarial defenses, providing new insights into why ensemble adversarial defenses are more effective.
2. Based on the theoretical analysis, the authors propose iGAT, which aims to narrow the subspace where all base classifiers misclassify inputs by allocating global adversarial examples to appropriate base classifiers and designing a regularization term for misclassified inputs.
3. The detailed experimental results demonstrate that the proposed methods significantly improve the adversarial robustness of ensemble adversarial defenses.


**Weaknesses:**

There is a lack of experiments with a small number of base classifiers. Will the proposed method still be effective when there are only a few base classifiers?

**Questions:**

Why is the robustness of ensemble defenses significantly lower than the robustness of state-of-the-art non-ensemble defenses? For example, the proposed method achieves 40.36% robustness under AutoAttack on the CIFAR10 dataset with a perturbation budget of 8/255. However, state-of-the-art non-ensemble adversarial training achieves 70.69% robustness under the same conditions (See RobustBench for detail).

**Limitations:**

The core theorem only addresses the simple case where there are only two MLP classifiers.

---

> ### Author Rebuttal · Authors · 2023-08-09
>
> We thank you for the positive comments and the accurate summary of our work and address the comments by grouping them into three categories: weaknesses, questions, and limitations.
>
> # Weaknesses
> * **On Weakness 1.** Encouraged by your comment, we have conducted a new experiment to examine performance improvement of iGAT over the two baselines of ADP and DVERGE, for an ensemble with only two base classifiers and the average combiner. The results are shown below, which will be included in the revised draft. The improvement is still satisfactory when using only two base classifiers.
>
> ---
> |Table 1. Results on ensembles of two base classifiers using the average combiner.|
> ---
> ---
> |||Natural (\%)|PGD (\%)|SH (\%)|AA (\%)|
> |---|---|---|---|---|---|
> |CIFAR10|ADP|73.02|31.94|33.17|25.72|
> ||iGAT$\_{ADP}$|**73.92**|**35.88**|**37.78**|**27.77**|
> |CIFAR100|DVERGE|55.72|21.39|28.26|16.97|
> ||iGAT$\_{DVERGE}$|**56.09**|**21.67**|**29.41**|**17.98**|
> ---
>
> # Questions
> * **Q1.** The main reason for this is the model size. Obviously, the bigger the model is the better their performance is when being trained with sufficient data. For instance, the top-performing models in   RobustBench [A] are WRN-70-16 of 267M, WRN-106-16 of at least 250M, and WRN-34-10 of 184M  [B]. According to the leaderboard of CIFAR10 data [C], the top-performing models are over 600M.  But the size of our ensemble model experimented in the paper is only 9M, containing 8 base ResNet-20 each of 1.1M.  We chose to experiment with such modest-sized models because the state-of-the-art adversarial defense works (see [14,23,30,42,50,51] in the submitted manuscript) that we compare with have all used the same. We adopted the same model architecture to focus on examining the effectiveness of the proposed enhancement approach.
>
> [A] Croce F et al. RobustBench: a standardized adversarial robustness benchmark, NeurIPS, 2021}
>
> [B] https://github.com/yaodongyu/TRADES}
>
> [C] https://paperswithcode.com/sota/image-classification-on-cifar-10
>
> # Limitations
>
> We appreciate the reviewer for pointing out this limitation.  We have actually made very promising progress after we submitted the paper. We have now managed to extend the theory to an arbitrary number of base MLPs, and plan to submit the new results to a suitable venue soon. Only for a matter of interest,  we briefly explain our extended proof sketch here. Suppose there are $M$ base models that can classify one sample, (e.g., $\\mathbf{x}\_i$) well and $N$ base models that classify the other sample (e.g., $\\mathbf{x}\_j$) well. For the max combiner, let $I\_M := \\{i_m\\}\_{m=1}^M$ be the indices of the $M$ base models and $I\_N := \\{i_n\\}\_{n=1}^N$ be the indices of the $N$ base models. Then, according to Eq. (62), given an index $i\_m \\in I\_M$, it has $\\max\_{c\\ne y\_i} \\max (h^{i\_m}\_c(\\mathbf{x}\_i), h^{i\_n}\_c(\\mathbf{x}\_i)) < h\_{y\_i}^{i\_m}(\\mathbf{x}\_i)$ for any $i\_n\\in I\_N$. This would lead to $0$ loss on $\\mathbf{x}\_i$. Likewise, given an index $i\_n\\in I\_N$, it has $\\max\_{c\\ne y\_j} \\max (h^{i\_n}\_c(\\mathbf{x}\_j), h^{i\_m}\_c(\\mathbf{x}\_j)) < h\_{y\_j}^{i\_n}(\\mathbf{x}\_j)$ for any $i\_m\\in I\_M$, which then give $0$ loss on $\\mathbf{x}\_j$. As our current proof shows one single classifier can have at least $1$ loss on the two samples $\\mathbf{x}\_i$ and $\mathbf{x}\_j$, the combination of more than two base models can perform better. For the average combiner, we will show that when $M= N$, the combination of more than two base models performs at least the same as single-branch models, and when $M\\ne N$, the ensemble model performs better.

---

> > ### Comment · Reviewer_XCW5 · 2023-08-13
> > **Thanks for the rebuttal**
> >
> > Thanks for providing the rebuttal. I will keep my original rating.

---

### Official Review · Reviewer_hPTz · 2023-07-06

**Soundness:** 2 fair
**Presentation:** 2 fair
**Contribution:** 3 good
**Rating:** 6
**Confidence:** 3

**Summary:**

1. This work aims at creating a theoretical framework to explain why an ensemble of MLPs can be more robust than the single components.
2. The assumptions under which the ensemble of MLPs have more robustness are given. The first set of conditions are on each individual model itself. The second set of assumption state that the performance of each classifier is acceptable in data points that are somehow close to points of other classes.
3. Under these assumptions, Theorem 4.1 states that an ensemble of two acceptable models always performs better than the individual ones.
4. Authors propose Interactive Global Adversarial Training to boost the robustness of an ensemble.


**Strengths:**

1. Experiments involve a wide variety of ensemble defense baselines and attacks.
2. Ablation studies are performed to test the effect of the different proposed mechanisms.
3. Assumptions in theoretical part are enumerated thoroughly.
4. Experimental setup is well described
5. The motivation to understand and enhance the adversarial robustness of ensembles is an interesting research direction.
6. Experimental results suggest that the proposed technique can indeed increase the robustness of existing ensemble training frameworks.


**Weaknesses:**

1. Theory and practice feel a bit disconnected. Theory works for 2 models, and only MLPs, and experiments are made with 8 Resnets.
2. Some assumptions in the theoretical part could be discussed in more detail to let the reader know if the requirements and assumptions could actually hold in practice or not.


**Questions:**

1. In Equation (5), what is the expectation over $h \in \mathcal{H}$? Are you computing the average error rate over all the family of hyothesis?
2. In Assumption 4.2 for MLPs, assumption 1 says "each activation output is i.i.d with expectation $\mu$". Is this reasonable ? What does this mean for the MLP ? Does this restrict the family of activation functions one can consider ?
3. In Assumption 4.2 for MLPs, what are the orders of magnitude of the constants $\tilde{\lambda}$ and $B$ ? How does this impact the bound on Equation (6) ?
4. Is Assumption 4.4 about "Acceptable Classifiers" always reasonable in practice ?
5. Why do you think that for CIFAR-10, the ensemble diversity framework that obtained the largest improvement from your proposed technique was ADP ? And why for CIFAR-100 it seems to be DVERGE ?
6. Why do you think that the iGAT parameters $\alpha$ and $\beta$ are so different for SoE and the rest of baselines ?
7. Which attack was used for the iGAT training added global adversarial loss?
8. How much time did training take ? Your implementation shows 2000 epochs were used, which seems huge for me. Does your method require more epochs than the baselines ?
Typos:
      Line 254: Theory 4.1  --> Theorem 4.1
      Model size column in Table 2, Appendix 3 seems to be wrong.



**Limitations:**

Yes, they have done.

---

> ### Author Rebuttal · Authors · 2023-08-09
>
> We thank you for the insightful comments and questions and address them below.
> # Weaknesses
> * **On Weakness 1.** We appreciate the reviewer for pointing this out.  By the time we submitted this paper, we were only able to develop the error theory for two base MLPs under those reported assumptions. But we have made very promising progress to improve it since. Now, we have managed to extend the theory to an arbitrary number of base MLPs with more relaxed assumptions and plan to submit the new results to a suitable venue soon.
>
> * **On Weakness 2.** We thank the reviewer for the very useful comment. We will include more discussions on our assumptions, also their practical feasibility and indication in the revised draft, based on what has been addressed in our reply to the questions below.
>
> # Questions
> * **Q1.** Yes, you are right.
>
> * **Q2.** We acknowledge that the assumption of independently having the same expectation $\mu$ for all the activation outputs is somewhat restricted.  But  one can consider the commonly used batch normalization techniques that independently control the mean and variance of an activation output (or input) so that different activators can present outputs of the same mean and variance.  Therefore, for MLPs with batch normalization, our assumption is reasonable. In addition, if we consider fixed input for the neural network but allow i.i.d neural network weights, the i.i.d assumption is reasonable. We include a related paper concerning the i.i.d weights and layer outputs/inputs [*].
>
> [*] Apollonio N  et al. Normal approximation of Random Gaussian Neural Networks. arXiv preprint arXiv:2307.04486, 2023.
>
> * **Q3.** We have investigated empirically the values of the network curvature $\tilde{\lambda}$ and the weights-related quantity $B$ using neural networks trained by the ADP defense techniques, and recorded the values below. Smaller values of $\tilde{\lambda}$  and $B$ indicate a looser upper bound in Eq. (6).  According to our Definition 4.3, a looser upper bound allows us to define an ambiguous pair containing two intra-class examples that are less close to each other, thus less challenging to classify.
>
> ---
> |Table 1. Magnitude of $\tilde{\lambda}$ and $B$|
> ---
> ---
> |$\tilde{\lambda}$| $B$|
> |---|---|
> |0.06|<0.01|
> ---
>
> * **Q4.**  The assumption of being an acceptable classifier is reasonable, and we believe that it can be met fairly easily by base classifiers of an adversarial ensemble model that is fairly well-trained. In our submitted supplementary material, we have already included some results to examine whether the base classifiers from the iGAT ensemble are acceptable (e.g., Tables 4 and 6).  To state some observations here,  the percentages for correctly classifying examples with part one of Assumption 4.4 satisfied are  $56\\%$ for iGAT$\_{ADP}$, $79\\%$ for iGAT$\_{DVERGE}$ and $50\\%$ for iGAT$\_{CLDL}$, where each percentage is averaged over the 8 base classifiers. These are greater than $42.5\\%$. Correspondingly, the averaged percentages for incorrectly classifying examples with part two of Assumption 4.4 satisfied are $73.12\\%$ for iGAT$\_{ADP}$, $80.19\\%$ for iGAT$\_{DVERGE}$ and $78.39\\%$ for iGAT$\_{CLDL}$, which are fairly high. Overall, for these examined cases, Assumption 4.4 is satisfied with a fairly high chance. We will enrich our discussion on this in the revised draft.
>
> * **Q5.** DVERGE performs proficiently on both CIFAR10 and CIFAR100. This can be credited to its approach of not only achieving adversarial diversity among base models but also ensuring that each sample is learned by multiple base models for stable classifications. Conversely, ADP's performance on CIFAR100 is not as robust as on CIFAR10. We aim to explain this by delving into its algorithm design. The novel aspect of ADP is its effort to maximize the prediction disparities among base models, thereby ensuring ensemble diversity. This implies that each model becomes proficient in classifying a subset of classes that other base models may struggle with. When the class count rises from 10 (in CIFAR10) to 100 (in CIFAR100), the inherent limitations of a single base model's capacity to efficiently classify a multitude of classes become evident. Consequently, the ADP approach, which discourages base models from learning overlapping classes, may not perform as effectively on datasets with a larger number of classes.
>
> * **Q6.** Both $\alpha$ and $\beta$ were tuned using grid search,  and their searching ranges were determined by the magnitude of losses used in the original ensemble methods. The loss construction of SoE differs from other methods quite a lot, which requires different searching ranges to match its magnitude and thus results in quite different values of $\alpha$ and $\beta$.
>
> * **Q7.** We used the PGD attack to generate adversarial training examples and to simulate global adversarial attacks, as it is the most commonly used by existing works.
>
> * **Q8.** We exemplify the training time by comparing the  ADP defense and its enhancement iGAT$\_{ADP}$.  Firstly, we have observed the training time required by updating the weights of ADP and iGAT$\_{ADP}$ using 1024 training examples 1000 times, which are  $50.431$ and $50.088$ in seconds. We can say that they take a similar time to train for the same number of iterations using the same amount of data.  Then, we have recorded the total number of training epochs required by both, which is $691$ epochs for ADP   while  $163$ epochs for  iGAT$\_{ADP}$. Based on these, we can conclude that  iGAT$\_{ADP}$ trains faster than ADP. In practice, the user will train their ensemble model using ADP first, and fine-tune it using iGAT$\_{ADP}$, which will require around $691 +193 <900$ epochs in total to obtain a good model. The epoch number 2000 is the maximum allowed epoch number that we set. The training terminates a lot earlier than that.

---

> > ### Comment · Reviewer_hPTz · 2023-08-20
> >
> > Thank you for your detailed answers. I will keep my original rating.

---

### Official Review · Reviewer_cK43 · 2023-07-12

**Soundness:** 3 good
**Presentation:** 3 good
**Contribution:** 3 good
**Rating:** 7
**Confidence:** 2

**Summary:**

This paper investigates enhancing the performance of ensemble-based adversarial defense methods against adversarial attacks. Considering the effectiveness of ensemble-based adversarial defense methods have been proven empirically while the understanding of their mechanism is rather limited, authors first propose a theory model to demonstrate the advantages of an ensemble of adversarially trained classifiers compared to individual ones. Guided by theoretical analysis, a novel method is developed to enhance the performance of existing ensemble-based defense methods through distributing different examples to different base classifiers. Experiments conducted on various datasets verify the superiority of the proposed method.

**Strengths:**

1. This paper proposes theoretical analysis on understanding the advantages of ensemble-based adversarial defense methods, which provides a new perspective to evaluate ensemble-based defense methods and hence could be helpful for future works
2. Based on theoretical understanding, a novel method is proposed to boost the performance of existing ensemble-based defense methods with reduced training time
3. Experiment results demonstrate the effectiveness of the proposed method

**Weaknesses:**

1. In experiments, authors apply the proposed method to enhance four state-of-the-art ensemble adversarial defense methods. However, in Table 1, only TOP 3 most enhanced performance are provided. It would be better if the performance related to all four ensemble defense methods can be provided
2. One feature of the proposed method is distributing different examples to different base classifiers. As author mentioned, this feature can make the proposed method bring reduced training time. It would be better if some experiments about training efficiency can be conducted

**Questions:**

1. Based on both Table 1 and Table 2, the improvement brought by the proposed method is not consistent among different ensemble-based defense methods. For example, for average combiner, the most significant performance improvement under AA is brought by the proposed method with ADP on CIFAR10 while with DVERGE on CIFAR100. Can authors provide more discussion about this phenomenon? Or, under which scenarios, the proposed method can bring most significant improvement?
2. Can authors provide the performance of all four methods in Table 1?

**Limitations:**

If possible, authors can add some discussion about the limitations of the proposed method.

---

> ### Author Rebuttal · Authors · 2023-08-09
>
> We thank you for the very positive comments and the accurate summary of our work. We address the comments by grouping them into three categories on questions, weaknesses, and limitations.
>
> # Answers
> * **Q1.** DVERGE performs proficiently on both CIFAR10 and CIFAR100. This can be credited to its approach of not only achieving adversarial diversity among base models but also ensuring that each sample is learned by multiple base models for stable classifications. Conversely, ADP's performance on CIFAR100 is not as robust as on CIFAR10. We aim to explain this by delving into its algorithm design. The novel aspect of ADP is its effort to maximize the prediction disparities among base models, thereby ensuring ensemble diversity. This implies that each model becomes proficient in classifying a subset of classes that other base models may struggle with. When the class count rises from 10 (in CIFAR10) to 100 (in CIFAR100), the inherent limitations of a single base model's capacity to efficiently classify a multitude of classes become evident. Consequently, the ADP approach, which discourages base models from learning overlapping classes, may not perform as effectively on datasets with a larger number of classes.
>
> * **Q2.** We have added results for all methods using the average combiner. See our reply to weakness 1.
>
> # Discussion about weaknesses
>
> * 1.To follow the suggestion, we have now collected results for the worst performing method (the 4th), which are iGAT$\_{SoE}$ for CIFAR10 and iGAT$\_{CLDL}$ for CIFAR100 (see below), using the average combiner. We will include these to the performance Table 1 in the revised draft, and will also collect results for these two methods using the max combiner.
>
> ---
> |Table 1. Added results of iGAT$\_{SoE}$ for CIFAR10 and iGAT$\_{CLDL}$ for CIFAR100.|
> ---
> ---
> |Average Combiner (\%)|||||||
> |---|---|---|---|---|---|---|
> |CIFAR10|SoE|**82.19**|38.54|37.59|**59.69**|32.68|
> ||iGAT$\_{SoE}$|81.05|**40.58**|**39.65**|57.91|**34.50**|
> |CIFAR100|CLDL|58.09|18.47|18.01|29.33|**15.52**|
> ||iGAT$\_{CLDL}$|**59.63**|**18.78**|**18.20**|**29.49**|14.36|
> ---
>
> * 2. The distributing feature reduces the training data size to $\frac{1}{N}$ for training each base classifier, where  $N$ is the number of base classifiers. This has the direct effect of improving training time.   We have added a simple experiment by comparing the training time required by a full version of iGAT$\_{ADP}$ and that by a modified version with this distributing feature removed. CIFAR10 data are used for Evaluation. The table below shows that using the distributing feature can slightly reduce the training time.
>
> ---
> |Table 2. Evaluation of training efficiency averagely at an iteration.|
> ---
> ---
> ||Time elapsed (s)|
> |---|---|
> |Without the distributing feature|5.63|
> |With the distributing feature|5.42|
> ---

---

> > ### Comment · Reviewer_cK43 · 2023-08-20
> > **Thank you for your reply**
> >
> > I have read authors' reply, which provides a more comprehensive aspect for the proposed method, especially for the limitations. Hence, I raised my score to 7.

---

### Official Review · Reviewer_Xydx · 2023-07-12

**Soundness:** 3 good
**Presentation:** 3 good
**Contribution:** 3 good
**Rating:** 5
**Confidence:** 4

**Summary:**

The ensemble strategy has gained popularity in the field of adversarial defense, wherein multiple base classifiers are trained cooperatively to protect against adversarial attacks. While empirical evidence supports its effectiveness, the underlying theoretical explanations for why an ensemble of adversarially trained classifiers outperforms individual classifiers remain elusive. This paper aims to bridge this gap by presenting a novel error theory dedicated to comprehending ensemble adversarial defense, providing provable reductions in the 0-1 loss on challenging example sets encountered in adversarial defense scenarios.

Building upon this theory, the authors propose an innovative approach called interactive global adversarial training (iGAT) to further enhance ensemble adversarial defense. iGAT incorporates two key components: (1) a probabilistic distributing rule that selectively assigns globally challenging adversarial examples to different base classifiers, and (2) a regularization term designed to address the most severe weaknesses of the base classifiers.

Evaluations conducted on CIFAR10 and CIFAR100 datasets demonstrate that iGAT significantly improves the performance of existing ensemble adversarial defense techniques. Under both white-box and black-box attacks, iGAT achieves performance gains ranging from 1% to 17%. These results highlight the efficacy of iGAT in bolstering the resilience of ensemble classifiers against adversarial attacks.

**Strengths:**

1. Novel Error Analysis: The paper presents a new error theory specifically dedicated to understanding the advantages of ensemble adversarial defense. This analysis fills a crucial gap in the literature by providing theoretical explanations for why an ensemble of adversarially trained classifiers is more robust than individual classifiers. By offering a deeper understanding of ensemble defense mechanisms, the paper contributes to the advancement of the field.

2. Targeted Solution Algorithm: In response to the theoretical insights gained from the error analysis, the paper proposes an innovative approach known as interactive global adversarial training (iGAT). This algorithmic solution addresses the shortcomings of existing ensemble adversarial defense techniques. It introduces a probabilistic distributing rule that selectively allocates globally challenging adversarial examples to different base classifiers, along with a regularization term that targets the most severe weaknesses of these classifiers. The proposed iGAT algorithm is tailored to enhance the performance and robustness of ensemble classifiers.

**Weaknesses:**

1. Limited Validation on Large-Scale Datasets: The paper's experimental evaluation focuses on the CIFAR10 and CIFAR100 datasets, which are relatively smaller in scale compared to datasets like ImageNet. The absence of validation on larger datasets limits the generalizability of the proposed iGAT algorithm and its effectiveness in real-world scenarios.

2. Limited Range of Attack Algorithms: The evaluation of the proposed defense technique primarily considers a limited range of attack algorithms without exploring a wider variety, including black-box attacks. While the performance improvements demonstrated against the evaluated attack algorithms are promising, it is important to assess the effectiveness of iGAT against a broader set of attack strategies to ensure its robustness and applicability in different adversarial settings.

3. Lack of Clear Explanation on the Differentiability of Hard Distributing Rule: The paper introduces a probabilistic distributing rule, referred to as the hard distributing rule, to assign adversarial examples to base classifiers. However, the paper does not provide a clear explanation regarding the differentiability of this rule. This omission raises concerns about the potential presence of obfuscated gradients, which could hinder the effective training of the ensemble classifiers and impact the overall performance of iGAT.

**Questions:**

Please provide results on ImageNet, and explain  the differentiability of Hard Distributing Rule

---

> ### Author Rebuttal · Authors · 2023-08-09
>
> We thank you for the insightful comments and address them regarding the following three aspects.
> # Answers
> * **Q1. On ImageNet Results:**
>
> We thank the reviewer for the suggestion. If allowed by time and the computing resource available to us, we would very much like to report the ImageNet results before the end of the rebuttal period, but unfortunately, we are not able to do so.  Using our computing resource of only one NVIDIA V100 GPU plus 8 CPU cores, it takes around 10-16 days to train one normal ResNet20 using the ImageNet data, and it takes even longer time to train a robust RestNet20 enhanced by an adversarial defense technique. Then, to train 8 such base ResNet20 and fine-tune them by the proposed adversarial ensemble enhancement, it will take more than 10 days $\\times$ 8 $\\times$ 1.2 > 3  months.  We are currently investigating computing resource availability and setting experiments for ImageNet, and hoping to obtain some ImageNet results for an ensemble with less than 8 ResNet20 in a shorter time.
>
> In general,  most of the state-of-the-art works on ensemble adversarial defense (e.g., those listed below) validate their models using CIFAR10 and CIFAR100 datasets, and some use simpler or even smaller datasets for digit recognition like the MNIST and SVHN datasets. This is why we have followed a similar evaluation setup.
>
> [A] Pang T et al. Improving adversarial robustness via promoting ensemble diversity, ICML, 2019.
>
> [B] Yang H et al. DVERGE: diversifying vulnerabilities for enhanced robust generation of ensembles, NeurIPS, 2020.
>
> [C] Yang Z  et al. Trs: Transferability reduced ensemble via promoting gradient diversity and model smoothness, NeurIPS, 2021.
>
> [D] Gao X and Xu C Z. MORA: Improving Ensemble Robustness Evaluation with Model Reweighing Attack, NeurIPS, 2022.
>
> [E] Cui S et al. Synergy-of-Experts: Collaborate to Improve Adversarial Robustness. NeurIPS, 2022.
>
> [F] Dbouk H and  Shanbhag N. On the Robustness of Randomized Ensembles to Adversarial Perturbations, ICML, 2023.
>
> * **Q2. On Differentiability of Hard Distributing Rule:**
>
>  We use the distributing rule to allocate different training samples to different base classifiers. The fine-tuning process by the proposed approach along with the classification loss and its gradient calculation start after the sample distributing process is completed. Thus, the training data used for fine-tuning each robust base classifier is fixed. There is no differentiation performed over these distributing rules.
>
> * **Q3. On Attack Algorithms:**
>
> Encouraged by the reviewer's comment regarding the limited range of attack algorithms, we have performed more experiments to test the effectiveness of iGAT against two more black-box attack strategies, using the CIFAR-10 dataset. These two attacks are two different black-box attacks that are time-efficient and commonly used, and we expect to use these two attacks to further compare the performance among ensemble defense models. Results show that in most cases the enhanced ensemble defense model has a better robustness against various black-box attacks.
>
> ---
> |Table 1. Results on extra black-box attacks.|
> ---
> ---
> ||Simple Attack (\%)| Bandit Attack (\%)|
> |---|---|---|
> |ADP|75.91|59.21|
> |iGAT$_{ADP}$|**79.43**|**64.55**|
> |DVERGE|79.43|63.80|
> |iGAT$_{DVERGE}$|**79.61**|**64.89**|
> |CLDL|76.82|63.80|
> |iGAT$_{CLDL}$|**78.84**|**65.25**|
> |SoE|**76.22**|66.10|
> |iGAT$_{SoE}$|75.18|**66.50**|
> ---

---

### Decision · Program_Chairs · 2023-09-21

**Decision:**

Accept (poster)

**Comment:**

This paper develops an error based theory for explaining ensemble adversarial defenses. Based on the theory, the authors propose interactive global adversarial training (iGAT) to further enhance the ensemble adversarial defense. The proposed approach obtains improvements on CIFAR10 and CIFAR100. Although some reviewers were concerned about missing evaluations on large scale datasets, all the reviewers are leaning towards accepting the paper. The authors should consider the reviewers' comments and prepare the camera-ready paper.